# High-dimensional Additive Gaussian Processes under Monotonicity Constraints

**Andrés F. López-Lopera**
CERAMATHS, UPHF
59313 Valenciennes, France
`andres.lopezlopera@uphf.fr`

**François Bachoc**
IMT, UMR5219 CNRS
Université Paul Sabatier
31062 Toulouse, France

**Olivier Roustant**
IMT, UMR5219 CNRS
INSA Toulouse
31077 Toulouse, France

## Abstract

We introduce an additive Gaussian process framework accounting for monotonicity constraints and scalable to high dimensions. Our contributions are threefold. First, we show that our framework enables to satisfy the constraints everywhere in the input space. We also show that more general componentwise linear inequality constraints can be handled similarly, such as componentwise convexity. Second, we propose the additive MaxMod algorithm for sequential dimension reduction. By sequentially maximizing a squared-norm criterion, MaxMod identifies the active input dimensions and refines the most important ones. This criterion can be computed explicitly at a linear cost. Finally, we provide open-source codes for our full framework. We demonstrate the performance and scalability of the methodology in several synthetic examples with hundreds of dimensions under monotonicity constraints as well as on a real-world flood application.

## 1 Introduction

**The framework of additive and constrained Gaussian processes.** In high dimension, many statistical regression models are based on additive structures of the form:

$$y(x_1, \cdots, x_d) = y_1(x_1) + \cdots + y_d(x_d). \tag{1}$$

Although such structures may lead to more "rigid" models than non-additive ones, they result in simple frameworks that easily scale in high dimensions [1, 2]. Generalized additive models (GAMs) [1] and additive Gaussian processes (GPs) [3, 4] are the most common models in a wide range of applications. The latter can also be seen as a generalization of GAMs that allow uncertainty quantification. As shown in [3, 4], additive GPs can significantly improve modeling efficiency and have major advantages for interpretability. Furthermore, in non-additive small-dimensional GP models, adding inequality constraints (in particular monotonicity which will be the main focus of this paper) leads to more realistic uncertainty quantification in learning from real data [5–10].

**Contributions.** Our contributions are threefold. **1)** We combine the additive and constrained frameworks to propose an additive constrained GP (cGP) prior. Our framework is based on a finite-dimensional representation involving one-dimensional knots for each active variable. The corresponding mode predictor can be computed and posterior realizations can be sampled, both in a scalable way to high dimension. **2)** We suggest the additive MaxMod algorithm for sequential dimension reduction. At each step, MaxMod either adds a new variable to the model or inserts a knot for an active one. This choice is made by maximizing the squared-norm modification of the mode function, for which we supply exact expressions with linear complexity. **3)** We provide open-source codes for our full framework. We demonstrate the performance and scalability of our methodology with numerical examples in dimension up to 1000 as well as in a real-world application in dimension 37. MaxMod identifies the most important input variables, with data size as low as $n = 2d$ in dimension $d$. It also yields accurate models satisfying the constraints everywhere on the input space.

36th Conference on Neural Information Processing Systems (NeurIPS 2022).

**Range of applicability.**   The computational bottleneck of cGPs is sampling from the posterior distribution. Here, it boils down to sampling a constrained Gaussian vector with dimension equal to the number of knots. This is done with Hamiltonian Monte Carlo (HMC) [11] which currently works with several hundreds of knots. Notice that MaxMod enables to minimize this number of knots.

Our framework is illustrated with monotonicity constraints and can be directly applied to other componentwise constraints such as componentwise convexity. These constraints should be linear and such that satisfying them on the knots is equivalent to satisfying them everywhere (see Section 3.2).

**Related literature.**   Additive GPs have been considered in [3, 4, 12–14], to name a few. The benefit of considering inequality constraints in (non-additive) GPs is demonstrated in [5–10, 15] and in application fields such as nuclear safety [5], geostatistics [6], tree distributions [7], econometrics [16], coastal flooding [17], and nuclear physics [18]. Note that there is a bridge between cGPs and (frequentist) splines-based regression with reproducing kernel Hilbert spaces (RKHS) [19, 20].

Our cGP model and MaxMod algorithm are extensions of the works in [5, 6] and [21] (respectively) to the additive case. Besides GPs, [22] provides a stochastic optimization framework with RKHS that yields regression with inequality constraints. [23] suggests a non-parametric model for non-negative functions; a problem related to that of monotonicity constraints. To the best of our knowledge, our framework is the first that enables to satisfy the constraints (especially monotonicity) everywhere and to scale to high dimension (up to one thousand in our experiments). In particular, the aforementioned cGP works are not applicable in these dimensions.

Furthermore, [22] does not satisfy the constraints everywhere, but only approximately and in expectation, and [23] does not directly address monotonicity constraints. For instance, with increasing constraints in dimension one, our framework provides posterior sample functions $h$ satisfying $h(u) \leq h(v)$ for *all* $u \leq v$. In contrast, the derivative of $h$, $h'(u)$, would be guaranteed to be non-negative only at some points $u$ in [15]. Also, $h'(u)$ would only be guaranteed to be above some strictly negative threshold, in expectation in [22].

Note that for unconstrained GP regression, the likelihood and conditional distribution are explicit but their computation cost scales as $\mathcal{O}(n^3)$, where $n$ is the size of the dataset. An active strand of literature then focuses on providing efficient numerical procedures [24, 25]. Because of the constraints, our work is complementary to this strand of literature. Indeed, with cGPs, even if $n$ is small, the numerical challenge of sampling from the posterior distribution remains, as discussed previously. The main indicator of complexity is thus the number of knots, rather than the dataset size.

The references [24, 25] provide convergence guarantees. At this stage, we do not provide convergence guarantees in our setting. To reach that goal, a possibility would be to extend the convergence proof of MaxMod given in [21] from the non-additive to the additive case. Two comments are in order on this. First, the proof techniques in [21] are different from these in [24, 25] (and also from standard asymptotic statistics settings, for instance [26]). Indeed, $n$ is fixed and the number of knots increases in [21], while $n$ increases in [24, 25]. Second, extending the proof of [21] to the additive case is not straightforward, since the various spaces of (non-additive) functions and their closures in [21] would have to be appropriately redefined for additive functions. Note also that [21] relies on a convergence proof in [27], that would need to be extended from the non-additive to the additive setting.

**Paper organization.**   Section 2 describes the additive GP framework. Sections 3 and 4 present our framework for additive cGPs and introduce the MaxMod algorithm. Section 5 provides the numerical experiments. Appendix A gathers the technical proofs as well as additional details and experiments.

## 2   Framework on additive Gaussian processes

In additive models, GP priors are placed here over the functions $y_1, \ldots, y_d$ in (1) [3, 4]. For $i = 1, \ldots, d$, let $\{Y_i(x_i); x_i \in [0, 1]\}$ be a zero-mean GP with covariance function (or kernel) $k_i$. Taking $Y_1, \ldots, Y_d$ as independent GPs, the process $\{Y(\boldsymbol{x}); \boldsymbol{x} \in [0, 1]^d\}$, with $\boldsymbol{x} = (x_1, \ldots, x_d)$, that results from the addition of $Y_1, \ldots, Y_d$, is also a GP and its kernel is given by

$$k(\boldsymbol{x}, \boldsymbol{x}') = k_1(x_1, x_1') + \cdots + k_d(x_d, x_d'). \tag{2}$$

In regression tasks, we often train the GP $Y$ to noisy data $(\boldsymbol{x}_\kappa, y_\kappa)_{1 \leq \kappa \leq n}$. We denote $x_i^{(\kappa)}$, for $\kappa = 1, \ldots, n$ and $i = 1, \ldots, d$, the element corresponding to the $i$-th input of $\boldsymbol{x}_\kappa$. By considering additive

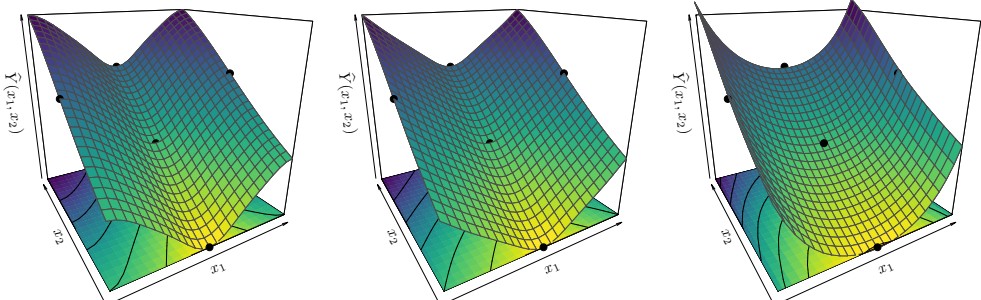

Figure 1: Additive GP predictions using (left) the unconstrained GP mean, (center) the cGP mode and (right) the cGP mean via HMC (see Section 3.3). The constrained model accounts for both componentwise convexity and monotonicity conditions along $x_1$ and $x_2$, respectively.

Gaussian noises $\varepsilon_\kappa \sim \mathcal{N}\left(0, \tau^2\right)$, with $\varepsilon_1, \cdots, \varepsilon_n$ assumed to be independent and independent of $Y$, then the conditional process $Y_n(\boldsymbol{x}) := Y(\boldsymbol{x})|\{Y(\boldsymbol{x}_1) + \varepsilon_1 = y_1, \ldots, Y(\boldsymbol{x}_n) + \varepsilon_n = y_n\}$ is GP-distributed with mean function and covariance function given by [3]

$$\mu(\boldsymbol{x}) = \boldsymbol{k}^\top(\boldsymbol{x})[\boldsymbol{K} + \tau^2 \boldsymbol{I}_n]^{-1} \boldsymbol{y}_n,$$
$$c(\boldsymbol{x}, \boldsymbol{x}') = k(\boldsymbol{x}, \boldsymbol{x}') - \boldsymbol{k}^\top(\boldsymbol{x})[\boldsymbol{K} + \tau^2 \boldsymbol{I}_n]^{-1} \boldsymbol{k}(\boldsymbol{x}),$$

where $\boldsymbol{y}_n = [y_1, \ldots, y_n]^\top$, $\boldsymbol{k}(\boldsymbol{x}) = \sum_{i=1}^d \boldsymbol{k}_i(x_i)$ with $\boldsymbol{k}_i(x_i) = [k_i(x_i, x_i^{(1)}), \ldots, k_i(x_i, x_i^{(n)})]^\top$, and $\boldsymbol{K} = \sum_{i=1}^d \boldsymbol{K}_i$ with $(\boldsymbol{K}_i)_{\kappa, \ell} = k_i(x_i^{(\kappa)}, x_i^{(\ell)})$ for $1 \leq \kappa, \ell \leq n$. The conditional mean $\mu$ and variance $v(\cdot) = c(\cdot, \cdot)$ are used for predictions and prediction errors, respectively.

**2D illustration.** Figure 1 shows the prediction of an additive GP modeling the function $(x_1, x_2) \mapsto 4(x_1 - 0.5)^2 + 2x_2$. The GP is trained with a squared exponential (SE) kernel, $k(\boldsymbol{x}, \boldsymbol{x}') = \sum_{i=1}^d \sigma_i^2 \exp(-(x_i - x_i')^2/2\ell_i^2)$ and with $(x_1, x_2)$: $(0.5, 0)$, $(0.5, 0.5)$, $(0.5, 1)$, $(0, 0.5)$, $(1, 0.5)$. The covariance parameters $\boldsymbol{\theta} = ((\sigma_1^2, \ell_1), (\sigma_2^2, \ell_2))$, corresponding to the variance and length-scale parameters (respectively), and the noise variance $\tau^2$ are estimated via maximum likelihood [28]. Although the resulting GP does preserve the additive condition, from Figure 1 (left) we can observe that the quality of the prediction will depend on the availability of data. In our example, we can observe that the GP model does not properly capture the convexity condition along $x_1$.

## 3 Contributions on additive Gaussian processes under inequality constraints

### 3.1 Finite-dimensional Gaussian process for fixed subdivisions

In order to satisfy the constraints everywhere (see the next subsection), we introduce the following finite-dimensional GP. For $i = 1, \ldots, d$ we consider a one-dimensional subdivision $\boldsymbol{S}_i$ (a finite subset of $[0, 1]$ composed of knots) with at least two knots at 0 and 1. Throughout Section 3, $\boldsymbol{S}_i$ is fixed, but its data-driven selection is studied in Section 4. If the number of knots of $\boldsymbol{S}_i$ is $m_i$, then the total number of knots is given by $m = m_1 + \cdots + m_d$. We let $\boldsymbol{S} = (\boldsymbol{S}_1, \ldots, \boldsymbol{S}_d)$. The finite-dimensional GP, denoted by $Y_{\boldsymbol{S}}(\boldsymbol{x})$, is written, for $\boldsymbol{x} \in [0, 1]^d$,

$$Y_{\boldsymbol{S}}(\boldsymbol{x}) = \sum_{i=1}^d Y_{i, \boldsymbol{S}_i}(x_i) = \sum_{i=1}^d \sum_{j=1}^{m_i} \xi_{i,j} \phi_{i,j}(x_i), \tag{3}$$

where $\xi_{i,j} = Y_i(t_{(j)}^{(\boldsymbol{S}_i)})$ with $Y_1, \ldots, Y_d$ GPs as in Section 2, and with $0 = t_{(1)}^{(\boldsymbol{S}_i)} < \cdots < t_{(m_i)}^{(\boldsymbol{S}_i)} = 1$ the knots in $\boldsymbol{S}_i$. We let $t_{(0)}^{(\boldsymbol{S}_i)} = -1$ and $t_{(m_i+1)}^{(\boldsymbol{S}_i)} = 2$ by convention, and $\phi_{i,j} = \phi_{t_{(j-1)}^{(\boldsymbol{S}_i)}, t_{(j)}^{(\boldsymbol{S}_i)}, t_{(j+1)}^{(\boldsymbol{S}_i)}} :$ $[0, 1] \to \mathbb{R}$ is the hat basis function centered at the knot $t_{(j)}^{(\boldsymbol{S}_i)}$ of $\boldsymbol{S}_i$. That is, for $u < v < w$, we let

$$\phi_{u,v,w}(t) = \begin{cases} \frac{1}{v-u}(t-u) & \text{for } u \leq t \leq v, \\ \frac{1}{w-v}(w-t) & \text{for } v \leq t \leq w, \\ 0 & \text{for } t \notin [u, w]. \end{cases} \tag{4}$$

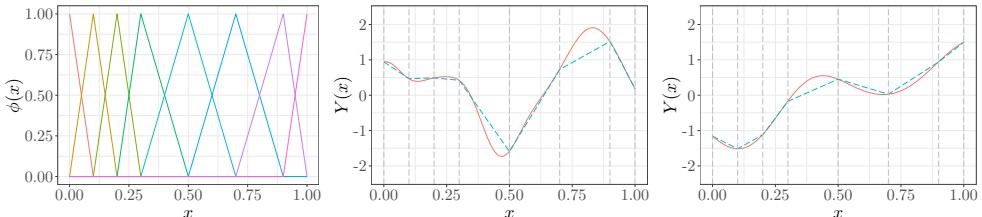

Figure 2: (left) 1D illustrations of the hat basis functions in (4) and (center and right) two random realizations from a standard GP (solid red line) and the corresponding finite dimensional GP (dashed blue line). For the latter, the location of the knots are plotted by vertical dashed lines.

Observe from (3) that, since $\xi_{i,j}$, for $i = 1, \ldots, d$ and $j = 1, \ldots, m_i$, are Gaussian distributed, then $Y_{i,\boldsymbol{S}_i}$ is a GP with kernel given by

$$\widetilde{k}_i(x_i, x_i') = \sum_{j=1}^{m_i} \sum_{\kappa=1}^{m_i} \phi_{i,j}(x_i)\phi_{i,\kappa}(x_i')k_i(t_{(j)}^{(\boldsymbol{S}_i)}, t_{(\kappa)}^{(\boldsymbol{S}_i)}). \tag{5}$$

Moreover, $Y_{\boldsymbol{S}}$ is a GP with kernel $\widetilde{k}(\boldsymbol{x}, \boldsymbol{x}') = \sum_{i=1}^{d} \widetilde{k}_i(x_i, x_i')$. Figure 2 shows random realizations from a standard GP and a finite-dimensional GP using a SE kernel with $\boldsymbol{\theta} = (\sigma^2, \ell) = (1, 0.1)$. We remark that we consider asymetric basis functions here, which is a novelty compared to [5, 6], that will allow MaxMod (Section 4) to promote non-equispaced designs. The additive finite-dimensional kernel in (5) also seems not to appear in the existing literature.

## 3.2 Satisfying inequality constraints

We consider the componentwise constraints $Y_{i,\boldsymbol{S}_i} \in \mathcal{E}_i$, $i = 1, \ldots, d$, where $\mathcal{E}_i$ is a one-dimensional function set. In line with [5, 6], we assume that there are convex sets $\mathcal{C}_i \subset \mathbb{R}^{m_i}$ such that

$$Y_{i,\boldsymbol{S}_i} \in \mathcal{E}_i \Leftrightarrow \boldsymbol{\xi}_i \in \mathcal{C}_i \tag{6}$$

where $\boldsymbol{\xi}_i = [\xi_{i,1}, \cdots, \xi_{i,m_i}]^\top$. Furthermore, for $i = 1, \ldots, d$, $\mathcal{C}_i$ is itself composed by $q_i$ linear inequalities:

$$\mathcal{C}_i = \left\{ \boldsymbol{c} \in \mathbb{R}^{m_i}; \forall \kappa = 1, \cdots, q_i : l_\kappa^{(i)} \leq \sum_{j=1}^{m} \lambda_{\kappa,j}^{(i)} c_j \leq u_\kappa^{(i)} \right\}, \tag{7}$$

where the $\lambda_{\kappa,j}^{(i)}$'s encode the linear operations and the $l_\kappa^{(i)}$'s and $u_\kappa^{(i)}$'s represent the lower and upper bounds. We write the constraints in (7) as $\boldsymbol{l}_i \leq \boldsymbol{\Lambda}_i \boldsymbol{c} \leq \boldsymbol{u}_i$.

Examples of such constraints are monotonicity and componentwise convexity. For instance, for the case where the function $y$ in (1) is non-decreasing with respect to each input, then $\mathcal{E}_i$ is the set of non-decreasing functions on $[0, 1]$ and $\mathcal{C}_i$ is given by

$$\mathcal{C}_i = \{\boldsymbol{c} \in \mathbb{R}^{m_i}; \forall j = 2, \cdots, m_i : c_j - c_{j-1} \geq 0\}. \tag{8}$$

Hence in this case, $Y_{\boldsymbol{S}}$ is monotonic on the entire $[0, 1]^d$ if and only if (by definition) each of its additive component $Y_{i,\boldsymbol{S}_i}$ is monotonic on $[0, 1]$, which happens if and only if the $m_i - 1$ constraints (8) are satisfied. We discuss the example of componentwise convexity in Appendix A.1.

## 3.3 Constrained GP predictions

We let $\boldsymbol{\Sigma}_i = k_i(\boldsymbol{S}_i, \boldsymbol{S}_i)$ be the $m_i \times m_i$ covariance matrix of $\boldsymbol{\xi}_i$. We consider $\boldsymbol{x}_1, \ldots, \boldsymbol{x}_n \in [0, 1]^d$ and write $\boldsymbol{\Phi}_i$ for the $n \times m_i$ matrix with element $(a, b)$ given by $\phi_{i,b}(\boldsymbol{x}_a)$. Then

$$\boldsymbol{Y}_n := [Y_{\boldsymbol{S}}(\boldsymbol{x}_1), \cdots, Y_{\boldsymbol{S}}(\boldsymbol{x}_n)]^\top = \sum_{i=1}^{d} \boldsymbol{\Phi}_i \boldsymbol{\xi}_i. \tag{9}$$

By considering noisy data $(\boldsymbol{x}_\kappa, y_\kappa)_{1 \leq \kappa \leq n}$, we have the regression conditions $\boldsymbol{Y}_n + \boldsymbol{\varepsilon}_n = \boldsymbol{y}_n$, where $\boldsymbol{\varepsilon}_n \sim \mathcal{N}(0, \tau^2 \boldsymbol{I}_n)$ and is independent from $\boldsymbol{\xi}_1, \ldots, \boldsymbol{\xi}_d$. Then given the observations and the constraints, the *maximum à posteriori* (MAP) estimate, also called the mode function, is given by

$$\widehat{Y}_{\boldsymbol{S}}(\boldsymbol{x}) = \sum_{i=1}^{d} \sum_{j=1}^{m_i} \widehat{\xi}_{i,j} \phi_{i,j}(x_i). \tag{10}$$

The vector $\widehat{\boldsymbol{\xi}} = [\widehat{\boldsymbol{\xi}}_1^\top, \ldots, \widehat{\boldsymbol{\xi}}_d^\top]^\top$ with $\widehat{\boldsymbol{\xi}}_i = [\widehat{\xi}_{i,1}, \ldots, \widehat{\xi}_{i,m_i}]^\top$ is the mode of the Gaussian distribution $\mathcal{N}(\boldsymbol{\mu}_c, \boldsymbol{\Sigma}_c)$ of the values at the knots conditionally to the observations and truncated from the constraints $\boldsymbol{l}_1 \leq \boldsymbol{\Lambda}_1 \boldsymbol{\xi}_1 \leq \boldsymbol{u}_1, \ldots, \boldsymbol{l}_d \leq \boldsymbol{\Lambda}_d \boldsymbol{\xi}_d \leq \boldsymbol{u}_d$ as in (7):

$$\widehat{\boldsymbol{\xi}} = \underset{\substack{\boldsymbol{c} = (\boldsymbol{c}_1^\top, \ldots, \boldsymbol{c}_d^\top)^\top \\ \boldsymbol{l}_i \leq \boldsymbol{\Lambda}_i \boldsymbol{c}_i \leq \boldsymbol{u}_i, i=1,\ldots,d}}{\operatorname{argmin}} (\boldsymbol{c} - \boldsymbol{\mu}_c)^\top \boldsymbol{\Sigma}_c^{-1} (\boldsymbol{c} - \boldsymbol{\mu}_c). \tag{11}$$

Above $\boldsymbol{\mu}_c = [\boldsymbol{\mu}_{c,1}^\top, \ldots, \boldsymbol{\mu}_{c,d}^\top]^\top$ is the $m \times 1$ vector with block $i$ given by

$$\boldsymbol{\mu}_{c,i} = \boldsymbol{\Sigma}_i \boldsymbol{\Phi}_i^\top \left[ \left( \sum_{p=1}^d \boldsymbol{\Phi}_p \boldsymbol{\Sigma}_p \boldsymbol{\Phi}_p^\top \right) + \tau^2 \boldsymbol{I}_n \right]^{-1} \boldsymbol{y}_n,$$

and $(\boldsymbol{\Sigma}_{c,i,j})_{i,j}$ is the $m \times m$ matrix with block $(i,j)$ given by

$$\boldsymbol{\Sigma}_{c,i,j} = \boldsymbol{1}_{i=j} \boldsymbol{\Sigma}_i - \boldsymbol{\Sigma}_i \boldsymbol{\Phi}_i^\top \left[ \left( \sum_{p=1}^d \boldsymbol{\Phi}_p \boldsymbol{\Sigma}_p \boldsymbol{\Phi}_p^\top \right) + \tau^2 \boldsymbol{I}_n \right]^{-1} \boldsymbol{\Phi}_j \boldsymbol{\Sigma}_j.$$

The expressions for $\boldsymbol{\mu}_c$ and $\boldsymbol{\Sigma}_c$ are obtained from the conditional formulas for Gaussian vectors [28]. Note that the matrix $\boldsymbol{\Sigma}_c$ is not block-diagonal, that is one cannot compute the posterior mode along each dimension separately. In Appendix A.2, based on the matrix inversion lemma [28, 29], efficient implementations to speed-up the computation of $\boldsymbol{\mu}_c$ and $\boldsymbol{\Sigma}_c$ are given when $m \ll n$.

Notice that the covariance parameters $\boldsymbol{\theta}$ of the kernel in (5) are estimated via maximum likelihood once the (unconstrained) additive GP framework is established. This is an intermediate step before solving the optimization problem in (11). Hence, given $\boldsymbol{\mu}_c, \boldsymbol{\Sigma}_c, \boldsymbol{\Lambda}_1, \ldots, \boldsymbol{\Lambda}_d, \boldsymbol{l}_1, \ldots, \boldsymbol{l}_d, \boldsymbol{u}_1, \ldots, \boldsymbol{u}_d,$ and $\boldsymbol{\theta}$, (11) is convex and can be solved via quadratic programming [5, 30]. To the best of our knowledge, the expression of the mode in (11) does not appear in the existing literature.

The cGP mode in (10) can be used as a point estimate of predictions. Since trajectories of $[\boldsymbol{\xi}_1^\top, \ldots, \boldsymbol{\xi}_d^\top]^\top$ conditionally on the observations and constraints can be generated via HMC [11], they can be used for uncertainty quantification. Furthermore, the mean of the HMC samples can be used for prediction purposes. Continuing the illustrative 2D example in Figure 1, we see the improvement brought by the mode function and the mean of conditional simulations, compared to the unconstrained additive GP model.

## 4 Additive MaxMod algorithm

### 4.1 Squared-norm criterion

Consider an additive cGP model that uses only a subset $\mathcal{J} \subseteq \{1, \ldots, d\}$ of active variables, with cardinality $|\mathcal{J}|$. We write its (vector of) subdivisions as $\boldsymbol{S} = (\boldsymbol{S}_i; i \in \mathcal{J})$. Its mode function $\widehat{Y}_{\boldsymbol{S}}$ is defined similarly as in (10), from $\mathbb{R}^{|\mathcal{J}|}$ to $\mathbb{R}$, by, for $\boldsymbol{x} = (x_i; i \in \mathcal{J})$,

$$\widehat{Y}_{\boldsymbol{S}}(\boldsymbol{x}) = \sum_{i \in \mathcal{J}} \sum_{j=1}^{m_i} \widehat{\xi}_{i,j} \phi_{i,j}(x_i). \tag{12}$$

Adding a new active variable $i^\star \notin \mathcal{J}$ to $\mathcal{J}$, and allocating it the base (minimal) subdivision $\boldsymbol{S}_{i^\star} = \{0, 1\}$ defines a new mode function $\widehat{Y}_{\boldsymbol{S}, i^\star} : \mathbb{R}^{|\mathcal{J}|+1} \to \mathbb{R}$. Adding a knot $t \in [0, 1]$ to the subdivision $\boldsymbol{S}_{i^\star}$ for $i^\star \in \mathcal{J}$ also defines a new mode function $\widehat{Y}_{\boldsymbol{S}, i^\star, t} : \mathbb{R}^{|\mathcal{J}|} \to \mathbb{R}$. Since a new variable or knot increases the computational cost of the model (optimization dimension for the mode computation and sampling dimension for generating conditional trajectories via HMC), it is key to quantify its contribution benefit. We measure this benefit by the squared-norm modification of the cGP mode

$$I_{\boldsymbol{S}, i^\star} = \int_{[0,1]^{|\mathcal{J}|+1}} \left( \widehat{Y}_{\boldsymbol{S}}(\boldsymbol{x}) - \widehat{Y}_{\boldsymbol{S}, i^\star}(\boldsymbol{x}) \right)^2 d\boldsymbol{x} \quad \text{for } i^\star \notin \mathcal{J}, \tag{13}$$

$$I_{\boldsymbol{S}, i^\star, t} = \int_{[0,1]^{|\mathcal{J}|}} \left( \widehat{Y}_{\boldsymbol{S}}(\boldsymbol{x}) - \widehat{Y}_{\boldsymbol{S}, i^\star, t}(\boldsymbol{x}) \right)^2 d\boldsymbol{x} \quad \text{for } i^\star \in \mathcal{J}, \tag{14}$$

where in the first case we see $\widehat{Y}_{\boldsymbol{S}}$ as a function of $|\mathcal{J}|+1$ variables that does not use variable $i^\star$.

## 4.2 Analytic expressions of the squared-norm criterion

**Adding a variable.** For a new variable $i^\star \notin \mathcal{J}$, the new mode function is

$$\widehat{Y}_{\boldsymbol{S},i^\star}(\boldsymbol{x}) = \sum_{i \in \mathcal{J}} \sum_{j=1}^{m_i} \widetilde{\xi}_{i,j} \phi_{i,j}(x_i) + \sum_{j=1}^{2} \widetilde{\xi}_{i^\star,j} \phi_{i^\star,j}(x_{i^\star}),$$

where $(\widetilde{\xi}_{i,j})_{i,j}$ and $(\widetilde{\xi}_{i^\star,j})_j$ follow from the optimization problem in dimension $\sum_{i \in \mathcal{J}} m_i + 2$ corresponding to (10) and (11). We let $\phi_{i^\star,1}(u) = 1 - u$ and $\phi_{i^\star,2}(u) = u$ for $u \in [0,1]$. Note that even though, for $i \in \mathcal{J}$ and $j = 1, \ldots, m_i$, $\widehat{\xi}_{i,j}$ and $\widetilde{\xi}_{i,j}$ correspond to the same basis function $\phi_{i,j}$, they are not equal in general as the new mode is reestimated with two more knots, which can modify the coefficients of all the knots. Next, we provide an analytic expression of the integral in (13).

**Proposition 1** (See proof in Appendix A.3). *We have*

$$I_{\boldsymbol{S},i^\star} = \sum_{i \in \mathcal{J}} \sum_{\substack{j,j'=1 \\ |j-j'| \leq 1}}^{m_i} \eta_{i,j} \eta_{i,j'} E_{j,j'}^{(\boldsymbol{S}_i)} - \sum_{i \in \mathcal{J}} \left( \sum_{j=1}^{m_i} \eta_{i,j} E_j^{(\boldsymbol{S}_i)} \right)^2 + \frac{\eta_{i^\star}^2}{12} + \left( \sum_{i \in \mathcal{J}} \sum_{j=1}^{m_i} \eta_{i,j} E_j^{(\boldsymbol{S}_i)} - \frac{\zeta_{i^\star}}{2} \right)^2,$$

*where* $\eta_{i,j} = \widehat{\xi}_{i,j} - \widetilde{\xi}_{i,j}$, $\eta_{i^\star} = \widetilde{\xi}_{i^\star,2} - \widetilde{\xi}_{i^\star,1}$, $\zeta_{i^\star} = \widetilde{\xi}_{i^\star,1} + \widetilde{\xi}_{i^\star,2}$, $E_j^{(\boldsymbol{S}_i)} := \int_0^1 \phi_{i,j}(t)dt$ *and* $E_{j,j'}^{(\boldsymbol{S}_i)} := \int_0^1 \phi_{i,j}(t)\phi_{i,j'}(t)dt$ *with explicit expressions in Lemma 1 in Appendix A.3. The matrices* $(E_{j,j'}^{(\boldsymbol{S}_i)})_{1 \leq j,j' \leq m_i}$ *are 1-band and the computational cost is linear with respect to* $m = \sum_{i \in \mathcal{J}} m_i$.

**Inserting a knot to a variable.** For a new $t$ added to $\boldsymbol{S}_{i^\star}$ with $i^\star \in \mathcal{J}$, the new mode function is

$$\widehat{Y}_{\boldsymbol{S},i^\star,t}(\boldsymbol{x}) = \sum_{i \in \mathcal{J}} \sum_{j=1}^{\widetilde{m}_i} \widetilde{\xi}_{i,j} \widetilde{\phi}_{i,j}(x_i),$$

where $\widetilde{m}_i = m_i$ for $i \neq i^\star$, $\widetilde{m}_{i^\star} = m_{i^\star} + 1$, $\widetilde{\phi}_{i,j} = \phi_{i,j}$ for $i \neq i^\star$, and $\widetilde{\phi}_{i^\star,j}$ is obtained from $\boldsymbol{S}_{i^\star} \cup \{t\}$ as in Lemma 1. As before, this follows from the optimization problem in dimension $\sum_{i \in \mathcal{J}} m_i + 1$ corresponding to (10) and (11). Next, we provide an analytic expression of (14).

**Proposition 2** (See proof in Appendix A.3). *For* $i \in \mathcal{J} \backslash \{i^\star\}$, *let* $\widetilde{\boldsymbol{S}}_i = \boldsymbol{S}_i$. *Let* $\widetilde{\boldsymbol{S}}_{i^\star} = \boldsymbol{S}_{i^\star} \cup \{t\}$. *Recall that the knots in* $\boldsymbol{S}_{i^\star}$ *are written* $0 = t_{(1)}^{(\boldsymbol{S}_{i^\star})} < \cdots < t_{(m_{i^\star})}^{(\boldsymbol{S}_{i^\star})} = 1$. *Let* $\nu \in \{1, \ldots, m_{i^\star} - 1\}$ *be such that* $t_{(\nu)}^{(\boldsymbol{S}_{i^\star})} < t < t_{(\nu+1)}^{(\boldsymbol{S}_{i^\star})}$. *Then, with a linear computational cost with respect to* $\widetilde{m} = \sum_{i \in \mathcal{J}} \widetilde{m}_i$, *we have*

$$I_{\boldsymbol{S},i^\star,t} = \sum_{i \in \mathcal{J}} \sum_{\substack{j,j'=1 \\ |j-j'| \leq 1}}^{\widetilde{m}_i} \bar{\eta}_{i,j} \bar{\eta}_{i,j'} E_{j,j'}^{(\widetilde{\boldsymbol{S}}_i)} - \sum_{i \in \mathcal{J}} \left( \sum_{j=1}^{\widetilde{m}_i} \bar{\eta}_{i,j} E_j^{(\widetilde{\boldsymbol{S}}_i)} \right)^2 + \left( \sum_{i \in \mathcal{J}} \sum_{j=1}^{\widetilde{m}_i} \bar{\eta}_{i,j} E_j^{(\widetilde{\boldsymbol{S}}_i)} \right)^2,$$

*where* $\bar{\eta}_{i,j} = \bar{\xi}_{i,j} - \widetilde{\xi}_{i,j}$, $E_j^{(\widetilde{\boldsymbol{S}}_i)}$ *and* $E_{j,j'}^{(\widetilde{\boldsymbol{S}}_i)}$ *are as in Proposition 1,* $\bar{\xi}_{i,j} = \widehat{\xi}_{i,j}$ *for* $i \neq i^\star$, $\bar{\xi}_{i^\star,j} = \widehat{\xi}_{i^\star,j}$ *for* $j \leq \nu$, $\bar{\xi}_{i^\star,j} = \widehat{\xi}_{i^\star,j-1}$ *for* $j \geq \nu + 2$, *and*

$$\bar{\xi}_{i^\star,\nu+1} = \widehat{\xi}_{i^\star,\nu} \frac{t_{(\nu+1)}^{(\boldsymbol{S}_{i^\star})} - t}{t_{(\nu+1)}^{(\boldsymbol{S}_{i^\star})} - t_{(\nu)}^{(\boldsymbol{S}_{i^\star})}} + \widehat{\xi}_{i^\star,\nu+1} \frac{t - t_{(\nu)}^{(\boldsymbol{S}_{i^\star})}}{t_{(\nu+1)}^{(\boldsymbol{S}_{i^\star})} - t_{(\nu)}^{(\boldsymbol{S}_{i^\star})}}.$$

## 4.3 The MaxMod algorithm

Algorithm 1 summarizes the routine of MaxMod. When considering inserting a knot $t$ to an active variable $i$, we promote space filling by adding a reward of the form $\Delta D(t, \boldsymbol{S}_i)$, where $\boldsymbol{S}_i$ is the current subdivision and where $D(t, \boldsymbol{S}_i)$ is the smallest distance from $t$ to an element of $\boldsymbol{S}_i$. When adding a new variable, we add a fixed reward $\Delta'$. Both $\Delta$ and $\Delta'$ are tuning parameters of the algorithm and allow to promote adding new variables over refining existing ones with new knots, or conversely. Step 3 in Algorithm 1 is performed by a grid search, and involves multiple optimizations for computing the new modes in (11), followed by applications of (14). Step 4 yields a single

---

**Algorithm 1** MaxMod for additive cGPs

---

**Input parameters:** $\Delta > 0$, $\Delta' > 0$, $d$.

1: Initialize MaxMod with the dimension in which the mode $\widehat{Y}_{\mathrm{MaxMod},0}$ maximizes the squared-norm.

**Sequential procedure:** For $m \in \mathbb{N}$, $m \geq 0$, do the following.

2: **for** $i = 1, \ldots, d$ **do**

3:      **if** the variable $i$ is already active **then** compute the optimal position of the new knot $t_i \in [0, 1]$ that maximizes $I_{\boldsymbol{S},i,t} + \Delta D(t, \boldsymbol{S}_i)$ over $t \in [0, 1]$, with $I_{\boldsymbol{S},i,t}$ as in (14). Denote the resulting mode as $\widehat{Y}_{\mathrm{MaxMod},m+1}^{(i)}$ and the resulting value of $I_{\boldsymbol{S},i,t}$ as $I_i$.

4:      **else** add two knots at the boundaries of the selected new active dimension, i.e. $(t_{i,1}, t_{i,2}) = (0, 1)$, and denote the resulting mode as $\widehat{Y}_{\mathrm{MaxMod},m+1}^{(i)}$ and the resulting value of (13) as $I_i$.

5: Choose the optimal decision $i^\star \in \{1, \ldots, d\}$ that maximizes the MaxMod criterion:

$$i^\star \in \mathrm{argmax}_{i \in \{1,\ldots,d\}} \left( I_i + \Delta \mathbf{1}_{i \in \mathcal{J}} D(t_i, \boldsymbol{S}_i) + \Delta' \mathbf{1}_{i \notin \mathcal{J}} \right).$$

6: Update knots and active variables and set new mode to $\widehat{Y}_{\mathrm{MaxMod},m+1} = \widehat{Y}_{\mathrm{MaxMod},m+1}^{(i^\star)}$.

---

optimization for computing the new mode in (11), followed by an application of (13). For each computation of a new mode, the covariance parameters of the kernels $k_i$, for $i \in \mathcal{J}$, are estimated. For faster implementations, the covariance parameters can be fixed throughout each pass corresponding to a fixed value of $m$ (Steps 2 to 6) or can be re-estimated every $T$ values of $m$, for some period $T \in \mathbb{N}$. Furthermore, steps 3 and 4 can be parallelized and computed in different clusters

## 5 Numerical experiments

Implementations of the additive cGP framework are based on the R package `lineqGPR` [31]. Experiments throughout this section are executed on an 11th Gen Intel(R) Core(TM) i5-1145G7 2.60GHz 1.50 GHz, 16 Gb RAM. Both R codes and notebooks to reproduce some of the numerical results are available in the Github repository: `https://github.com/anfelopera/lineqGPR`.

As training data, we use random Latin hypercube designs (LHDs). Note that using them is not necessary for the implementation of our methodology, but it is recommended when the user is able to design the experiments. Indeed, [32] shows that LH sampling reduces estimation variances compared to simple random sampling, especially for additive functions. Thus, using LHDs enables us to choose a minimal design size $n = 2d$, corresponding to the number of parameters of the chosen additive GP kernel: an additive Matérn 5/2 kernel with one variance parameter $\sigma_i^2$ and one characteristic-length parameter $\ell_i$ per dimension [3]. We denote $\boldsymbol{\theta} = ((\sigma_1^2, \ell_1), \cdots, (\sigma_d^2, \ell_d))$. For other applications where $n \in [d + 1, 2d[$, we may consider the kernel structure in [4] with only one global variance parameter, at the cost of more restricted cGP models. Finally, in the examples that require a comparison with non-additive GP models, we use common GP settings: maximin LHDs and $n = 10d$. The larger value of $n$ accounts for the fact that additivity is not a prior information of standard GPs.

We compare the quality of the GP predictions in terms of the $Q^2$ criterion on unobserved data. It is defined as $Q^2 = 1 - \mathrm{SMSE}$, where SMSE is the standardized mean squared error [28]. For noise-free data, $Q^2$ is equal to one if predictions are exactly equal to the test data and lower than one otherwise.

### 5.1 Additive Gaussian processes with monotonicity constraints

#### 5.1.1 Monotonicity in 5D

We start by an example in small dimensions, in order to compare with non-additive constrained GPs which do not scale to high dimensions. We thus consider the additive function given by

$$y(\boldsymbol{x}) = \arctan(5x_1) + \arctan(2x_2) + x_3 + 2x_4^2 + \frac{2}{1 + \exp\{-10(x_5 - \frac{1}{2})\}}, \qquad (15)$$

with $\boldsymbol{x} \in [0, 1]^5$. Observe that $y$ is non-decreasing with respect to all its input variables. We evaluate $y$ on a maximin LHD over $[0, 1]^5$ at $n = 50$ locations using [33]. In this example, as explained in the introduction of the section, we have chosen a maximin LHD rather than a random LHD, and

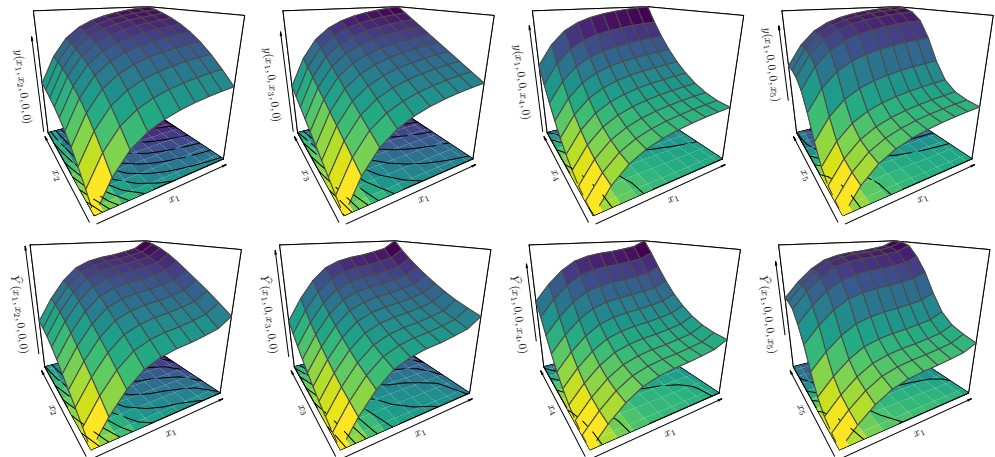

Figure 3: Additive GP under monotonicity constraints in 5D with $n = 10d$. 2D projections of the true profiles and the constrained GP mean predictions are shown in the first and second row, respectively.

Table 1: Results (mean $\pm$ one standard deviation over 10 replicates) on the monotonic example in Section 5.1.2 with $n = 2d$. Both computational cost and quality of the cGP predictions (mode and mean) are assessed. For the computation of the cGP mean, $10^3$ ($^\dagger 50$) HMC samples are used.

| $d$ | $m$ | CPU Time $[s]$ | | $Q^2$ [%] | | |
|---|---|---|---|---|---|---|
| | | cGP mode | cGP mean | GP mean | cGP mode | cGP mean |
| 10 | 50 | $0.1 \pm 0.1$ | $0.1 \pm 0.1$ | $82.3 \pm 6.2$ | $83.8 \pm 4.2$ | $\mathbf{88.1 \pm 1.7}$ |
| 100 | 500 | $0.4 \pm 0.1$ | $5.2 \pm 0.5$ | $89.8 \pm 1.6$ | $90.7 \pm 1.4$ | $\mathbf{91.5 \pm 1.3}$ |
| 250 | 1250 | $4.2 \pm 0.7$ | $132.3 \pm 26.3$ | $91.7 \pm 0.8$ | $92.9 \pm 0.6$ | $\mathbf{93.4 \pm 0.6}$ |
| 500 | 2500 | $37.0 \pm 11.4$ | $^\dagger 156.9 \pm 40.5$ | $92.5 \pm 0.6$ | $93.8 \pm 0.5$ | $^\dagger \mathbf{94.3 \pm 0.5}$ |
| 1000 | 5000 | $262.4 \pm 35.8$ | $^\dagger 10454.3 \pm 3399.3$ | $92.6 \pm 0.3$ | $94.6 \pm 0.2$ | $^\dagger \mathbf{95.1 \pm 0.2}$ |

$n = 10d$ rather than $n = 2d$, because we also consider non-additive GPs for which these settings are recommended. We fix 20 knots per dimension, leading to a total of $m = 100$ knots.

Figure 3 shows the additive cGP mean prediction under monotonicity constraints considering $10^4$ HMC samples. Our framework leads to improvements on both CPU time and quality of predictions compared to the non-additive cGP in [5]. Due to computational limitations, the non-additive cGP is performed with a reduced but tractable number of knots per dimension set to $m_1 = m_2 = m_4 = 4$, $m_3 = 2$ and $m_5 = 6$ , for a total of $m = 768$ knots. With this setup, and considering a factorial design with 11 test points per dimension, the additive cGP yields $Q^2 = 99.8\%$, an absolute improvement of 1.3% compared to the non-additive cGP. In term of the CPU times, the computation of the cGP mode and cGP mean using the additive framework are obtained in 6s and 0.8s (respectively), a significant speed-up compared to the non-additive cGP that required 28.9s and 24.3 minutes.

### 5.1.2 Monotonicity in hundreds of dimensions

For testing the additive cGP in high dimensions, we consider the target function used in [21]:

$$y(\boldsymbol{x}) = \sum_{i=1}^{d} \arctan\left(5\left[1 - \frac{i}{d+1}\right]x_i\right), \tag{16}$$

with $\boldsymbol{x} \in [0,1]^d$. This function exhibits decreasing growth rates as the index $i$ increases. For different values of $d$, we assess GPs with and without constraints. We fix $\boldsymbol{\theta} = (\sigma_i^2, \ell_i)_{1 \leq i \leq d} = (1,2)$. Although $\boldsymbol{\theta}$ can be estimated our focus here is to assess both computational cost and quality of cGP predictors. The cGP mean is obtained by averaging $10^3$ HMC samples. We set 5 knots per dimension.

Table 1 summarizes the CPU times and $Q^2$ values of the cGP predictors. The $Q^2$ criterion is computed considering $10^5$ evaluations of $y$ based on a LHD fixed for all the experiments. Results are shown over 10 replicates with different random LHDs. From Table 1, it can be observed that the cGPs lead

Table 2: $Q^2$ Performance of the MaxMod algorithm for the example in Section 5.2.1 with $n = 10D$.

| $D$ | $d$ | active dimensions | knots per dimension | $Q^2 (\widetilde{Y}_{\text{MaxMod}})$ [%] | $Q^2 (\widehat{Y}_{\text{MaxMod}})$ [%] |
|---|---|---|---|---|---|
| | 2 | $(1, 2)$ | $(4, 3)$ | 99.5 | **99.8** |
| 10 | 3 | $(1, 2, 3)$ | $(5, 5, 3)$ | 97.8 | **99.8** |
| | 5 | $(1, 2, 3, 4, 5)$ | $(4, 4, 4, 3, 2)$ | 91.4 | **99.8** |
| | 2 | $(1, 2)$ | $(5, 3)$ | 99.7 | **99.8** |
| 20 | 3 | $(1, 2, 3)$ | $(4, 4, 3)$ | 99.0 | **99.9** |
| | 5 | $(1, 2, 3, 4, 5)$ | $(5, 4, 3, 3, 2)$ | 96.0 | **99.7** |

to prediction improvements, with $Q^2$ (median) increments between 1.7-5.8%. Although the cGP mean predictor provides $Q^2$ values above 88%, it becomes expensive when $d \geq 500$. On the other hand, the cGP mode yields a trade-off between computational cost and quality of prediction.

## 5.2 MaxMod algorithm

### 5.2.1 Dimension reduction illustration

We test the capability of MaxMod to account for dimension reduction considering the function in (16). In addition to $(x_1, \ldots, x_d)$, we include $D - d$ virtual variables, indexed as $(x_{d+1}, \ldots, x_D)$, which will compose the subset of inactive dimensions. For any combination of $D \in \{10, 20\}$ and $d \in \{2, 3, 5\}$, we apply Algorithm 1 to find the true $d$ active dimensions. For each value of $D$, we evaluate $f$ at a maximin LHD with $n = 10D$. Similarly to Section 5.1.1 where we compare to non-additive GPs, we have used the common GP settings rather than the ones used here for additive GPs (random LHD, $n = 2d$). We compare the accuracy of two modes:

- $\widehat{Y}_{\text{MaxMod}}$: the mode resulting from the additive cGP and additive MaxMod.

- $\widetilde{Y}_{\text{MaxMod}}$: the mode resulting from the non-additive cGP in [5] with equispaced one-dimensional knots but where the number of knots per dimension is the same as for $\widehat{Y}_{\text{MaxMod}}$.

Table 2 shows that MaxMod correctly identifies the $d$ dimensions that are actually active where the most variable ones have been refined with more knots. There, MaxMod is considered to converge if the squared-norm criterion is smaller than $\epsilon = 5 \times 10^{-4}$. In terms of the $Q^2$ criterion, $\widehat{Y}_{\text{MaxMod}}$ leads to more accurate results compared to $\widetilde{Y}_{\text{MaxMod}}$, with $Q^2 \geq 99.7\%$ in all the cases.

### 5.2.2 Real application: flood study of the Vienne river

The database contains a flood study conducted by the French multinational electric utility company EDF in the Vienne river [34]. It is composed of $N = 2 \times 10^4$ simulations of an output $H$ representing the water level and 37 inputs depending on: a value of flow upstream, data on the geometry of the bed, and Strickler friction coefficients (see further details in Appendix A.5.1). This database was generated as a random design, assuming that the inputs are independent and follow specific probability distributions. In order to apply our framework, we use componentwise quantile transformations to make the inputs uniform. Such transformations do not modify the additivity and monotonicity properties, but may increase non-linearities. With expert knowledge, it is possible to identify that $H$ is decreasing along the first 24 input dimensions and increasing along dimension 37. The behavior along inputs 25 to 36 is not clear. Although underlying functions modeling the flood phenomena may be non additive, a previous sensitivity analysis in [34], done with the full database, has shown that the additive assumption is realistic here. Moreover, it has shown that inputs 11, 35 and 37 explain most of the variance. We aim at obtaining similar conclusions with a fraction of the simulation budget, using the information of additivity and monotonocity constraints with respect to dimensions 1 to 24 and 37.

In this experiment, MaxMod is applied for 10 replicates with random training sets and 10 iterations. The settings described at the beginning of Section 5 are followed as much as possible. As the database is fixed, we cannot directly use a random LHD of size $n = 2d$. Thus, we pick the closest subset of size $n$ (measured by the Euclidean norm). The remaining data are used for testing the cGPs. Figure 4 shows the choice of MaxMod per iteration and for the 10 replicates (adding or refining variables).

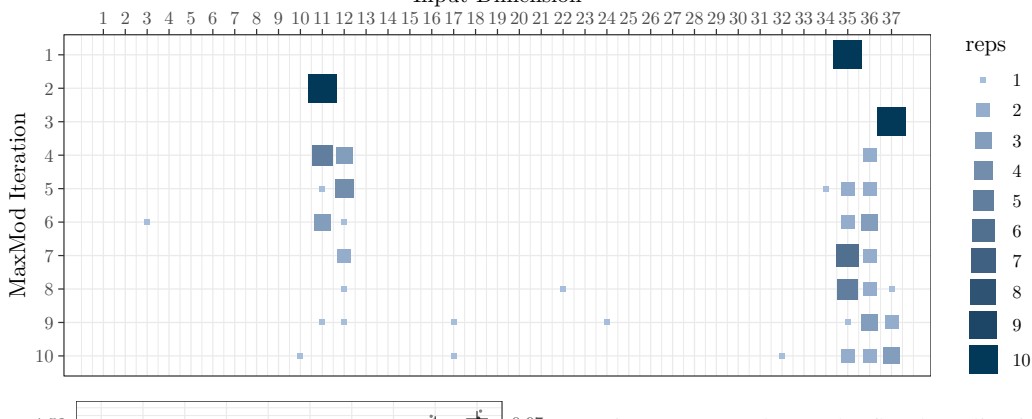

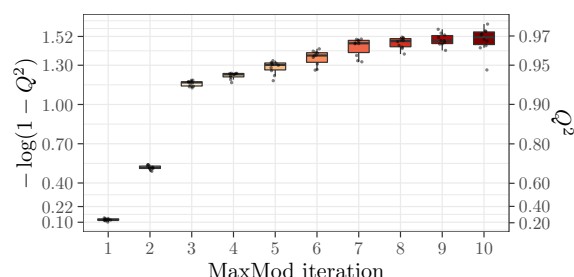

Figure 4: Results on the flood application in Section 5.2.2 with $n = 2d = 74$. The panels show: (top) the choice made by Max-Mod per iteration and (left) $Q^2$ boxplots per iteration of MaxMod. Results are computed over 10 random replicates. For the first panel, a bigger and darker square implies a more repeated choice.

The first three iterations of the algorithm show that MaxMod activates mainly dimensions 11, 35 and 37. In the subsequent iterations, MaxMod either refines the these dimensions or activates dimensions 12 and 36. In terms of the quality of predictions (Figure 4), MaxMod leads to $Q^2 > 0.92$ after the first three iterations.

Similar results are obtained for $n = 3d$ and $n = 4d$ (see Appendix A.5.2). MaxMod indeed activates dimensions 11, 35 and 37 in the first three iterations and refines them mainly in subsequent iterations. This allows to conclude that MaxMod correctly identifies the most relevant input dimensions and that accurate predictions are obtained once those dimensions are activated.

## Acknowledgments and Disclosure of Funding

This research was conducted within the frame of the ANR GAP Project (ANR-21-CE40-0007) and the consortium in Applied Mathematics CIROQUO, gathering partners in technological research and academia in the development of advanced methods for Computer Experiments (doi:10.5281/zenodo.6581217). We thank Bertrand Iooss and EDF R&D/LNHE for providing the Mascaret test case and Sébastien Petit who has performed the computations on this model.

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
