# A  Appendix

## A.1  Satisfying inequality constraints everywhere for componentwise convexity

For componentwise convexity, $\mathcal{E}_i$ in (6) is the set of one-dimensional convex functions and $\mathcal{C}_i$ is given by, with $(t_1, \ldots, t_{m_i}) = (t_{(1)}^{(\boldsymbol{S}_i)}, \ldots, t_{(m_i)}^{(\boldsymbol{S}_i)})$,

$$\mathcal{C}_i = \left\{ \boldsymbol{c} \in \mathbb{R}^{m_i}; \forall j = 3, \cdots, m_i : \frac{c_j - c_{j-1}}{t_j - t_{j-1}} \geq \frac{c_{j-1} - c_{j-2}}{t_{j-1} - t_{j-2}} \right\}. \tag{17}$$

We can see that, for each $i \in \{1, \ldots, d\}$ and each $\boldsymbol{x}_{-i} \in [0, 1]^{d-1}$, the one-dimensional cut $u \in [0, 1] \mapsto Y_{\boldsymbol{S}}(u, \boldsymbol{x}_{-i})$ (where only the input $i$ is varying) is convex if and only if each additive component $Y_{i, \boldsymbol{S}_i}$ is convex on $[0, 1]$. This happens if and only if the $m_i - 2$ inequality constraints in (17) are satisfied.

## A.2  Speed-up of numerical implementation

**Notation.** The expression (9) can be rewritten in the matrix form $Y_n := \sum_{i=1}^d \boldsymbol{\Phi}_i \boldsymbol{\xi}_i = \boldsymbol{\Psi} \boldsymbol{\xi}$ with $\boldsymbol{\Psi} = [\boldsymbol{\Phi}_1, \ldots, \boldsymbol{\Phi}_d]$ an $n \times m$ matrix and $\boldsymbol{\xi} = [\boldsymbol{\xi}_1^\top, \ldots, \boldsymbol{\xi}_d^\top]^\top$ an $m \times 1$ vector. With this notation, we have the following expressions for $\boldsymbol{\mu}_c$ and $\boldsymbol{\Sigma}_c$ (see Section 3.3):

$$\boldsymbol{\mu}_c = \boldsymbol{\Sigma} \boldsymbol{\Psi}^\top \boldsymbol{C}^{-1} \boldsymbol{y}_n,$$
$$\boldsymbol{\Sigma}_c = \boldsymbol{\Sigma} - \boldsymbol{\Sigma} \boldsymbol{\Psi}^\top \boldsymbol{C}^{-1} \boldsymbol{\Psi} \boldsymbol{\Sigma},$$

where $\boldsymbol{C} = \boldsymbol{\Psi} \boldsymbol{\Sigma} \boldsymbol{\Psi}^\top + \tau^2 \boldsymbol{I}_n$ and $\boldsymbol{\Sigma} = \mathrm{bdiag}(\boldsymbol{\Sigma}_1, \ldots, \boldsymbol{\Sigma}_d)$ is an $m \times m$ block diagonal matrix.

The computation of $\boldsymbol{C}^{-1}$, required in GP predictions and estimation of the covariance parameters via maximum likelihood, can be performed more efficiently when $m \ll n$ using properties of matrices [28, 29]. Next, we detail how the computational complexity of $\boldsymbol{C}^{-1}$ can be reduced from $\mathcal{O}(n^3)$ to $\mathcal{O}(m^3)$. We also provide an efficient computation of the determinant $|\boldsymbol{C}|$ that is required in covariance parameter estimation.

**Computation of $\boldsymbol{C}^{-1}$.** To avoid numerical instability issues, we first rewrite $\boldsymbol{C}$ in terms of the Cholesky decomposition $\boldsymbol{\Sigma} = \boldsymbol{L} \boldsymbol{L}^\top$. Here, $\boldsymbol{L}$ is an $m \times m$ block lower-triangular matrix given by $\boldsymbol{L} = \mathrm{bdiag}(\boldsymbol{L}_1, \cdots, \boldsymbol{L}_d)$ with $\boldsymbol{L}_i$ the Cholesky decomposition of $\boldsymbol{\Sigma}_i$ for $i = 1, \ldots, d$. Thus, $\boldsymbol{C}^{-1} = [(\boldsymbol{\Psi} \boldsymbol{L}) \boldsymbol{I}_m (\boldsymbol{\Psi} \boldsymbol{L})^\top + \tau^2 \boldsymbol{I}_n]^{-1}$. Now, by applying the matrix inversion lemma (see, e.g., [28], Appendix A.3), we obtain:

$$\boldsymbol{C}^{-1} = [(\boldsymbol{\Psi} \boldsymbol{L}) \boldsymbol{I}_m (\boldsymbol{\Psi} \boldsymbol{L})^\top + \tau^2 \boldsymbol{I}_n]^{-1} = \tau^{-2} [\boldsymbol{I}_n - \boldsymbol{\Psi} \boldsymbol{L} (\tau^2 \boldsymbol{I}_m + \boldsymbol{L}^\top \boldsymbol{\Psi}^\top \boldsymbol{\Psi} \boldsymbol{L})^{-1} \boldsymbol{L}^\top \boldsymbol{\Psi}^\top].$$

We need to compute now the inversion of the $m \times m$ matrix $\boldsymbol{A} = \tau^2 \boldsymbol{I}_m + \boldsymbol{L}^\top \boldsymbol{\Psi}^\top \boldsymbol{\Psi} \boldsymbol{L}$. Let $\widetilde{\boldsymbol{L}}$ be the Cholesky decomposition of $\boldsymbol{A}$. Denote $\boldsymbol{P} = \boldsymbol{\Psi} \boldsymbol{L}$. Then $\boldsymbol{C}^{-1}$ is given by

$$\boldsymbol{C}^{-1} = \tau^{-2} [\boldsymbol{I}_n - (\widetilde{\boldsymbol{L}}^{-1} \boldsymbol{P}^\top)^\top \widetilde{\boldsymbol{L}}^{-1} \boldsymbol{P}^\top].$$

Since $\widetilde{\boldsymbol{L}}$ is a lower triangular matrix, then $\widetilde{\boldsymbol{L}} \boldsymbol{M} = \boldsymbol{P}^\top$ can be sequentially solved in $\boldsymbol{M}$.

In addition to reducing complexity to $\mathcal{O}(m^3)$ for $m \ll n$, some of the intermediate steps here can be parallelized. For instance, the computation of the $d$ Cholesky matrices $\boldsymbol{L}_i$ with $m_i < m$.

**Computation of $|\boldsymbol{C}|$.** From [28] (Appendix A.3), and considering the Cholesky decomposition $\boldsymbol{\Sigma} = \boldsymbol{L} \boldsymbol{L}^\top$, we have that the determinant $|\boldsymbol{C}|$ is given by

$$|\boldsymbol{C}| = |(\boldsymbol{\Psi} \boldsymbol{L}) \boldsymbol{I}_m (\boldsymbol{\Psi} \boldsymbol{L})^\top + \tau^2 \boldsymbol{I}_n| = \tau^{2(n-m)} |\tau^2 \boldsymbol{I}_m + \boldsymbol{L}^\top \boldsymbol{\Psi}^\top \boldsymbol{\Psi} \boldsymbol{L}| = \tau^{2(n-m)} |\widetilde{\boldsymbol{L}}|^2,$$

where $|\widetilde{\boldsymbol{L}}| = \prod_{j=1}^m \widetilde{L}_{j,j}$ with $\widetilde{L}_{j,j}$ the element associated to the $j$-th row and $j$-th column of $\widetilde{\boldsymbol{L}}$.

## A.3  Squared-norm criterion

The next lemma is elementary to show and its second part is also given in [21].

**Lemma 1.** *Consider a subdivision $S = \{u_1, \ldots, u_m\}$ and write its ordered knots as $0 = u_{(1)} < \cdots < u_{(m)} = 1$. For $j = 1, \ldots, m$, write the hat basis function $\phi_j$ as $\phi_{u_{(j-1)}, u_{(j)}, u_{(j+1)}}$ in (4), with the conventions $u_{(0)} = -1$ and $u_{(m+1)} = 2$. For $j = 1, \ldots, m$, we have*

$$E_j^{(S)} := \int_0^1 \phi_j(t)dt = \begin{cases} \dfrac{u_{(j+1)}}{2} & \text{if } j = 1 \\[2mm] \dfrac{1 - u_{(j-1)}}{2} & \text{if } j = m \\[2mm] \dfrac{u_{(j+1)} - u_{(j-1)}}{2} & \text{if } j \in \{2, \ldots, m-1\} \end{cases}.$$

*For $j, j' = 1, \ldots, m$, we have*

$$E_{j,j'}^{(S)} := \int_0^1 \phi_j(t)\phi_{j'}(t)dt = \begin{cases} \dfrac{u_{(j+1)} - u_{(j)}}{3} & \text{if } j = j' = 1 \\[2mm] \dfrac{u_{(j)} - u_{(j-1)}}{3} & \text{if } j = j' = m \\[2mm] \dfrac{u_{(j+1)} - u_{(j-1)}}{3} & \text{if } j = j' \in \{2, \ldots, m-1\} \\[2mm] \dfrac{u_{(j+1)} - u_{(j)}}{6} & \text{if } j' = j+1 \\[2mm] \dfrac{u_{(j)} - u_{(j-1)}}{6} & \text{if } j' = j-1 \\[2mm] 0 & \text{if } |j - j'| \geq 2 \end{cases}.$$

*Proof of Proposition 1.* From a probabilistic point of view, assuming that the input variables are random variables, i.e. $(X_i, i \in \mathcal{J} \cup \{i^\star\})$ are uniformly distributed on $[0,1]$ and independent, then (13) can be rewritten as an expectation,

$$I_{S,i^\star} = \mathbb{E}\left( \left( \sum_{i \in \mathcal{J}} \widehat{Y}_i(X_i) - \sum_{i \in \mathcal{J} \cup \{i\}} \widehat{Y}_{i^\star,i}(X_i) \right)^2 \right),$$

where, for $i \in \mathcal{J}$, $\widehat{Y}_i = \sum_{j=1}^{m_i} \widehat{\xi}_{i,j} \phi_{i,j}$, and for $i \in \mathcal{J} \cup \{i^\star\}$, $\widehat{Y}_{i^\star,i} = \sum_{j=1}^{m_i} \widetilde{\xi}_{i,j} \phi_{i,j}$ (with $m_{i^\star} = 2$). Then

$$I_{S,i^\star} = \text{Var}\left( \sum_{i \in \mathcal{J}} \widehat{Y}_i(X_i) - \sum_{i \in \mathcal{J} \cup \{i\}} \widehat{Y}_{i^\star,i}(X_i) \right) + \mathbb{E}^2 \left( \sum_{i \in \mathcal{J}} \widehat{Y}_i(X_i) - \sum_{i \in \mathcal{J} \cup \{i\}} \widehat{Y}_{i^\star,i}(X_i) \right)$$

$$= \sum_{i \in \mathcal{J}} \text{Var}\left( \sum_{j=1}^{m_i} (\widehat{\xi}_{i,j} - \widetilde{\xi}_{i,j})\phi_{i,j}(X_i) \right) + \text{Var}\left( \sum_{j=1}^{2} \widetilde{\xi}_{i^\star,j} \phi_{i^\star,j}(X_{i^\star}) \right) \tag{18}$$

$$+ \left( \sum_{i \in \mathcal{J}} \sum_{j=1}^{m_i} (\widehat{\xi}_{i,j} - \widetilde{\xi}_{i,j})\mathbb{E}(\phi_{i,j}(X_i)) - \sum_{j=1}^{2} \widetilde{\xi}_{i^\star,j}\mathbb{E}(\phi_{i^\star,j}(X_{i^\star})) \right)^2.$$

Recall $\eta_{i,j} = \widehat{\xi}_{i,j} - \widetilde{\xi}_{i,j}$. We have for $i \in \mathcal{J}$,

$$\text{Var}\left( \sum_{j=1}^{m_i} \eta_{i,j}\phi_{i,j}(X_i) \right) = \mathbb{E}\left( \left( \sum_{j=1}^{m_i} \eta_{i,j}\phi_{i,j}(X_i) \right)^2 \right) - \left( \mathbb{E}\left( \sum_{j=1}^{m_i} \eta_{i,j}\phi_{i,j}(X_i) \right) \right)^2$$

$$= \sum_{j,j'=1}^{m_i} \eta_{i,j}\eta_{i,j'} E_{j,j'}^{(S_i)} - \left( \sum_{j=1}^{m_i} \eta_{i,j} E_j^{(S_i)} \right)^2, \tag{19}$$

with the notation of Lemma 1. To compute the term relative to dimension $i^\star$ in (18), recall that, if $x$ belongs to the support $[0,1]$ of $X_{i^\star}$, then $\phi_{i^\star,1}(x) = 1 - x$ and $\phi_{i^\star,2}(x) = x$. Hence,

$$\text{Var}\left( \sum_{j=1}^{2} \widetilde{\xi}_{i^\star,j}\phi_{i^\star,j}(X_{i^\star}) \right) = \text{Var}(\widetilde{\xi}_{i^\star,1}(1 - X_{i^\star}) + \widetilde{\xi}_{i^\star,2}X_{i^\star}) = \frac{(\widetilde{\xi}_{i^\star,2} - \widetilde{\xi}_{i^\star,1})^2}{12}. \tag{20}$$

Using (19) and (20) in (18), and observing that $\mathbb{E}(\phi_{i^\star,j}(X_{i^\star})) = 1/2$ for $j = 1, 2$, concludes the proof. $\qquad\square$

Table 3: $Q^2$ performance of MaxMod for the example in Appendix A.4 with $d = 3$, $D = 10$ and $n = 10D$. The active dimensions are displayed in the order they have been activated by MaxMod.

| $\lambda$ | Sobol index $(x_{D-1}, x_D)$ | active dimensions | knots per dimension | $Q^2 \left(\widehat{Y}_{\text{MaxMod}}\right)$ [%] |
|---|---|---|---|---|
| 0 | $1.7 \times 10^{-5}$ | (2, 1, 3) | (5, 5, 3) | 99.8 |
| 0.5 | 0.02 | (2, 1, 3, 10, 9) | (5, 5, 3, 2, 2) | 99.2 |
| 1 | 0.08 | (1, 2, 3, 9, 10, 5) | (5, 4, 3, 2, 2) | 97.6 |
| 1.5 | 0.15 | (1, 2, 3, 10, 9) | (5, 4, 3, 2, 2) | 95.5 |
| 1.7 | 0.18 | (1, 2, 10, 3, 9, 5) | (5, 4, 2, 3, 2, 2) | 94.7 |

*Proof of Proposition 2.* As in the proof of Proposition 1, we write

$$I_{\boldsymbol{S}, i^\star, t} = \mathbb{E}\left( \left( \sum_{i \in \mathcal{J}} \widehat{Y}_i(X_i) - \sum_{i \in \mathcal{J}} \widehat{Y}_{i^\star, t, i}(X_i) \right)^2 \right),$$

where, for $i \in \mathcal{J}$, $\widehat{Y}_i = \sum_{j=1}^{m_i} \widehat{\xi}_{i,j} \phi_{i,j}$, and $\widehat{Y}_{i^\star, t, i} = \sum_{j=1}^{\widetilde{m}_i} \widetilde{\xi}_{i,j} \widetilde{\phi}_{i,j}$, where $\widetilde{\phi}_{i,j} = \phi_{i,j}$ for $i \neq i^\star$ and where $\widetilde{\phi}_{i^\star, j}$ is defined as in $\phi_j$ in Lemma 1 from the subdivision $\widetilde{\boldsymbol{S}}_{i^\star} = \boldsymbol{S}_i \cup \{t\}$.

We express $\widehat{Y}_{i^\star}$ from the current subdivision $\boldsymbol{S}_{i^\star}$ to the refined subdivision $\widetilde{\boldsymbol{S}}_{i^\star}$, as in Proposition SM2.1 in [21], which yields

$$\widehat{Y}_{i^\star} = \sum_{j=1}^{\widetilde{m}_{i^\star}} \bar{\xi}_{i^\star, j} \widetilde{\phi}_{i^\star, j}.$$

Then we can carry out the same computations as in the proof of Proposition 1 (as if we had $\widetilde{\xi}_{i^\star, j} = 0$ for $j = 1, 2$ in that proof) to conclude, also using Lemma 1. $\qquad \square$

### A.4 Robustness of MaxMod in the presence of non-additive functions

To enrich the discussion on the robustness of MaxMod in the presence of a non-additive component, we performed a new experiment with a target function given by

$$y(\boldsymbol{x}) = \sum_{i=1}^d \arctan\left( 5\left[1 - \frac{i}{d+1}\right] x_i \right) + \lambda x_{D-1} x_D,$$

with $d = 3$ and $D = 10$. Here, $\lambda \geq 0$ is a parameter that controls the influence of the non-additive contribution. Observe that, as $\lambda$ increases, the influence of $x_{D-1}$ and $x_D$ also increases.

We repeat the experiment in Section 5.2.1 for $\lambda = 0, 0.5, 1, 1.5, 1.7$ and for $n = 10D$ (value also used in Table 2). The values of $\lambda$ correspond to percentages of variance explained by the interaction term (Sobol index). For the Sobol analysis, we have used the R package `sensitivity` [35]. From Table 3, we observe that MaxMod properly activates dimensions $(x_1, x_2, x_3, x_9, x_{10})$ in the first iterations while preserving accurate $Q^2$ values. For $\lambda > 2$, corresponding to a Sobol index larger than 20% for the interaction term, the $Q^2$ performance of MaxMod decreases. This is expected since the additive cGP is not able to capture the non-additive behavior.

### A.5 Real application: flood study of the Vienne river (France)

#### A.5.1 Database description

The database consists of numerical simulations using the software Mascaret [34], which is a 1-dimensional free surface flow modeling industrial solver based on the Saint-Venant equations [36, 37]. It is composed of an output $H$ representing the water level and 37 inputs depending on a value of flow upstream that is disturbed by a value $dQ$, on data of the geometry of the bed that are disturbed by modifying the gradients of quantities $dZ_{ref}$, and on Strickler friction coefficients ($cf_2$ for the major bed and $cf_1$ for the minor bed). More precisely, the inputs correspond to:

- 12 Strickler coefficients corresponding to $cf_1$, denoted as $X_1, \ldots, X_{12}$, whose distributions are uniform over $[20, 40]$;
- 12 Strickler coefficients corresponding to $cf_2$, denoted as $X_{13}, \ldots, X_{24}$, whose distributions are uniform over $[10, 30]$;
- 12 $dZ_{ref}$ gradient perturbations, denoted as $X_{25}, \ldots, X_{36}$, with standard normal distributions truncated on $[-3, 3]$;
- and 1 upstream flow disturbance value $dQ$, denoted as $X_{37}$, with a centered normal distribution with standard deviation $\sigma = 50$ and truncated over $[-150, 150]$.

In [34], the laws (either uniform or truncated Gaussian) of the input parameters have been chosen arbitrarily, according to the empirical distributions observed during experimental campaigns. Moreover, these random variables have been assumed independent.

Developments in Section 4.2 assume that the input variables are independent and uniformly distributed. To account for laws different from the uniform one (e.g. the truncated Gaussian law), new analytic expressions of the squared-norm criterion imply more technical developments that are not provided in this work. However, the expectations in Appendix A.3 can still be approximated via Monte Carlo. In our numerical experiments, we preferred to apply a quantile transformation of the input space for having independent uniformly distributed random variables $X_1, \ldots, X_{37}$ on $[0, 1]^{37}$. It can be shown that this transformation preserves the monotonicity constraints and additive structure.

### A.5.2 Additional results

Figures 5 and 6 show the additional results discussed in Section 5.2.2 for $n = 3d$ and $n = 4d$.

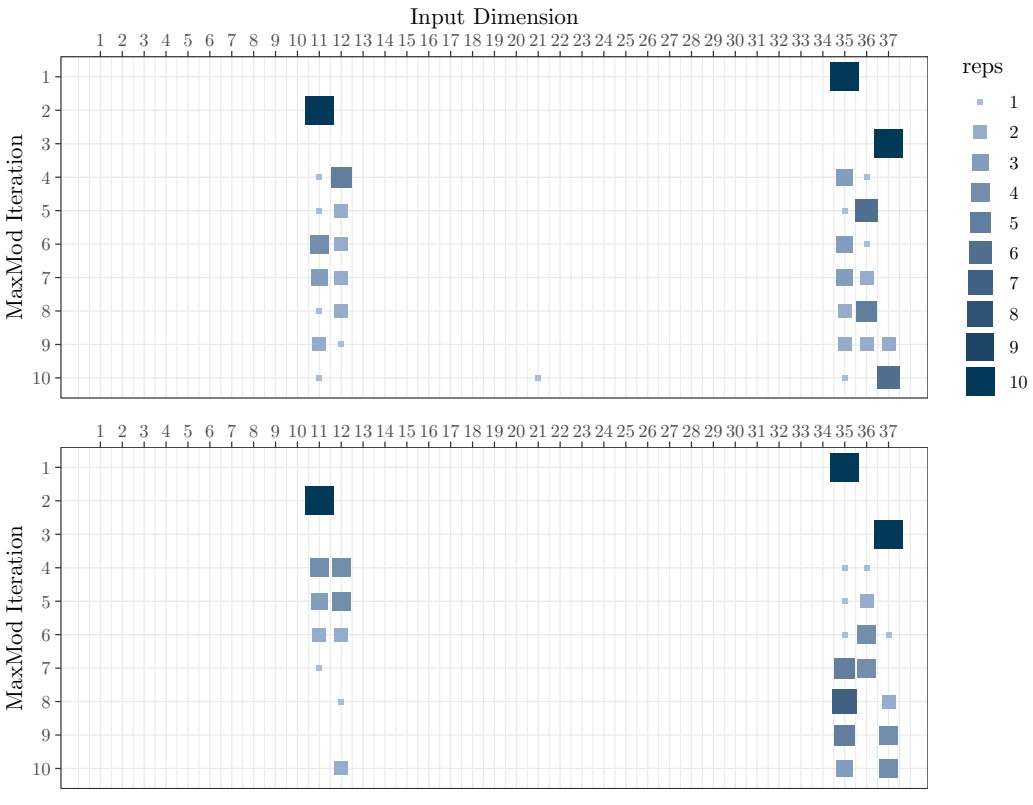

Figure 5: The choice made by MaxMod for the ten replicates considering the flood application in Section 5.2.2 with (top) $n = 3d = 111$ and (bottom) $n = 4d = 148$.

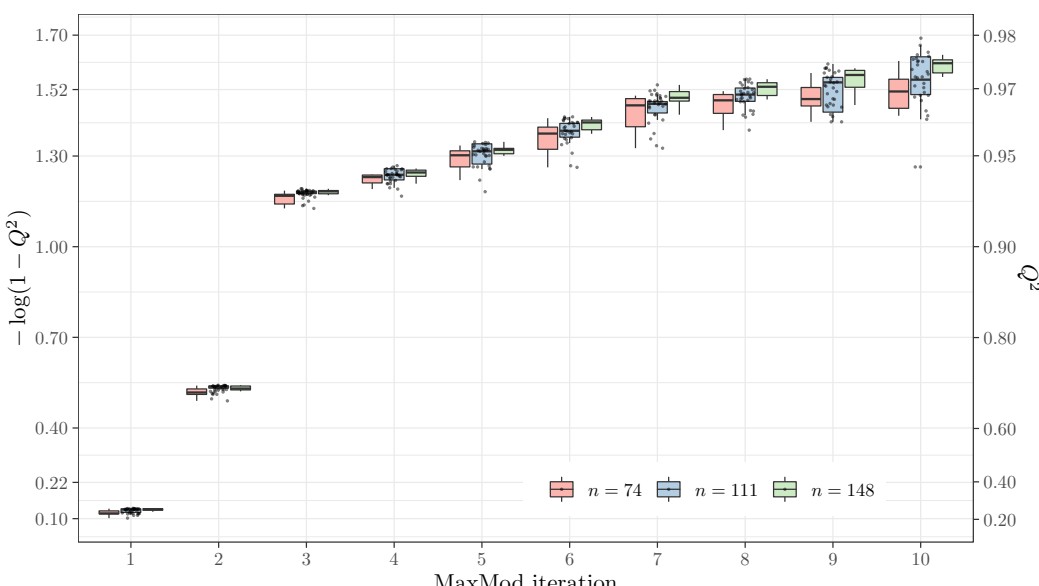

Figure 6: $Q^2$ boxplots per iteration of MaxMod for the flood application in Section 5.2.2. Results are shown for $n = 2d, 3d, 4d$.