# OpenReview forum: "High-dimensional Additive Gaussian Processes under Monotonicity Constraints"
_NeurIPS.cc/2022/Conference — NeurIPS 2022 Accept_

### Official Review · Reviewer_cQK3 · 2022-07-05

**Rating:** 7
**Confidence:** 4
**Soundness:** 3 good
**Presentation:** 3 good
**Contribution:** 2 fair

**Summary:**

The authors consider the problem of regression tasks with an additive Gaussian process prior and linear inequality constraint. The authors propose a finite dimensional approximation to the Gaussian process as a linear combination of triangular basis functions with Gaussian weights. The weights are then estimated via solving a quadratic program (for MAP estimation) or approximately sampled with HMC to handle the inequality constraints. Additionally, the authors consider the problem of variable selection in this model, and propose a forward selection method based on the difference between the posterior mode with and without the inclusion of a particular variable (or knot). The authors demonstrate their method for inference and variable selection on synthetic data, that is generated to be additive and monotonic along certain dimensions. Finally, the authors apply the method to a flood dataset generated by a utility company via simulation.

**Questions:**

1. (This is a comment) I think it would be useful to include a figure showing a.) the hat basis functions, b.) a sample from a 1-dimensional Gaussian process prior with a given kernel (e.g. Matérn 5/2) as well as the sample (using the same realization of Gaussian noise) modified to the finite dimensional model (eqn 5) with a handful of knots.

2. (Lines 96-97) You refer to a function as being monotonic on $[0,1]^d$. I don’t think this is a well-defined notion (since $[0,1]^d$ isn’t totally ordered). What do you mean by this? I assume you mean that restricted to each component it is monotonic, in which case the first equivalence you state is really just a definition. This should be made clear.

3. Equation 8 seems more restrictive than the original formulation in which $\mathcal{C}_i$ need only be convex. Are they somehow equivalent? If not, it would be best to initially introduce $\mathcal{C}_i$ as a set defined by finitely many linear inequality to avoid confusion.

4. Notationally, the first equality in (eqn 10) seems to suggest that the MAP estimate can be computed by computing a MAP estimate along each dimension separately and combining these. Is this what is intended? I don’t see why this would be true (and in equation 11 this does not seem to be the case). If it is, could you please give some explanation as to why? If not, consider changing the notation slightly.

5. (Minor, line 116) “et” -> “and”

6. (Minor) Is the information benefit directly tied to ideas from information theory? If not, it may be worth considering renaming.

7. (Minor, Algorithm 1, 5) “D” -> “d”

8. In the flood dataset experiments, you selected your training set via finding a subset as close as possible to a LHD (250-251). Would findings have differed if you had instead selected training data uniformly at random? Similarly, what if $n$ were somewhat larger than $2d$? In particular, is it important to the practical application of the method that the covariate design be something like a LHD? I didn’t see where an assumption like this was made in the discussion of the method, but it seems all of the experiments used this.

9. How robust is the proposed MaxMod algorithm in the presence of model mis-specification? For example, if the function being modelled has non-additive components, can this lead to issues with variable selection?

**Limitations:**

There is some discussion of limitations in the paper. I think it would be nice to see a synthetic experiment where the assumptions of the model are not precisely true (e.g. the function used to generate the data has some, perhaps small, additive component). I expect most real-world data is not exactly additive, and it would be nice to see how robust the proposed method is to small mis-specifications.

**Strengths And Weaknesses:**

### Strengths
- The scope of the paper is well-defined and of interest to those modeling with Gaussian processes. Incorporating prior knowledge such as inequality constraints into these models seems to be an important task.
- The forward selection algorithm (MaxMod) for selecting which dimensions to include seems appealing for high-dimensional datasets, and the experiments seem to suggest that it can be effective.
- The writing is generally quite clear.

### Weaknesses
- I am concerned that the contribution regarding extension of existing work to handle high-dimensional, additive models is over-claimed. It looks to me like the method considered is very similar to the references [5,6], applying the prior works approach to each dimension (separately).If there is a major distinction that I have missed between the approaches, I would appreciate if the authors could highlight this more explicitly. Otherwise, I think this contribution should be contextualized a bit more clearly relative to existing work, particularly in section 3 where it isn't entirely clear what is new and what is a based directly on existing methods.

---

> ### Author Response · Authors · 2022-07-29
> **Response to reviewer cQK3 (Part 1/2)**
>
> We thank the reviewer for the careful reading of the paper, the overall positive evaluation, and the constructive suggestions, in particular about the assessment of the MaxMod algorithm with model misspecification. Below are our responses to the reviewer's comments and questions.
>
> ---
>
> **Minor changes:**
> - We corrected the typos pointed out by the reviewer and carefully double-checked the paper.
> - We changed the term "information benefit" to "contribution benefit".
> - We generated figures with 1D illustrations of the hat basis functions, and samples from a GP and a finite-dimensional GP (without constraints). We will add them in Section 3.1.
> - We agree that the intermediate equality in Eq. 10 is misleading, although mathematically correct. We decided to remove it. Indeed, it is not correct to compute a MAP estimate along each dimension separately, since the GP values across different dimensions have non-zero posterior covariances.
>
> **Novel contributions related to constrained GPs compared to [5,6]:**
> Although our main contributions are already summarized in the introduction, we agree that adding remarks throughout Section 3 to highlight the differences between our work and the ones in [5,6] will improve the discussion. Some of them will be related to:
> - The construction of the asymmetric hat basis functions (compared to the symmetric ones considered in [5,6]). It allows the MaxMod algorithm to insert knots promoting non-equispaced designs.
> - The novel additive kernel considering the non-centered hat basis functions.
> - The new prediction formulas for the additive case (see Eq. (11) and expressions below). We will highlight that our framework is not a simple application of the priors from [5,6] to each dimension since the values at the knots (called the Gaussian weights by the reviewer) at all the dimensions, conditioned to the observations and constraints, are mutually dependent. Thus, they need to be jointly estimated.
>
> **On the definition of the set of linear inequality constraints:**
> As stated in Section 3.2 we consider componentwise constraints. Thus a multivariate function is non-decreasing iff, by definition, it is non-decreasing w.r.t. each component (lines 93-95). Indeed, the first equivalence at lines 96-97 is just a definition. We will clarify this in the text.
>
> Eq. (8) is indeed more restrictive than having a general convex set $\mathcal{C}_i$. Nevertheless, only (8) makes our implementation possible, yielding numerical optimization with a finite number of linear constraints. To make this clearer, in Section 3.2, we will directly consider a convex set $\mathcal{C}_i$ of the form (8).
>
> **On the choice of LHD:**
> In a preliminary experiment on the flood application, we indeed considered a random selection of the training sets. We observed similar results to the ones presented in the paper for $n>4d$. For $n=2d,3d$, the methodology led to higher variability in the order selection of the input variables and the $Q^2$ results. We still noted that MaxMod properly activated the most relevant dimensions in the first iterations of the algorithm.
>
> Hence, using LHDs is not necessary for the implementation of our methodology. However, we recommend it when the user is able to design the experiments. Indeed, as mentioned at the beginning of Section 5, the theoretical benefits of LH sampling for additive functions have been demonstrated in [23]. In practice, as seen above for the flood application, using LHDs can reduce the variability in the order selection of the input variables and improve prediction performance. In the paper, we will enrich the discussion of the impact of the experimental design.
>
> *(to be continued in the next comment)*

---

> > ### Author Response · Authors · 2022-07-29
> > **Response to reviewer cQK3 (Part 2/2)**
> >
> > *(follow-up of Part 1)*
> >
> > **Robustness of MaxMod in the presence of model misspecification:**
> > We must remark that the underlying function in the flood application is indeed non-additive with weak interactions between the input variables (see further details in [25]).
> >
> > To enrich the discussion on the robustness of MaxMod in the presence of a non-additive component, we performed a new experiment with a function given by
> > $$
> > y(\textbf{x})=\sum_{i=1}^{d}\arctan\left(5\bigg[1-\frac{i}{d+1}\bigg] x_i\right)+\lambda x_{D-1}x_D,
> > $$
> > with $d=3$ and $D=10$. $\lambda\geq0$ is a parameter that controls the influence of the non-additive contribution. Observe that, as $\lambda$ increases, the influence of the input variables $x_{D-1},x_D$ also increases.
> >
> > After running the experiments for $\lambda=0,0.5,1,1.5,1.7$ (values chosen so that the Sobol indices for the input dimensions $x_9$ and $x_{10}$ are smaller than 1/5), and for $n=10D$ (value also used in Table 2), we observed that MaxMod properly activates dimensions ($x_1,x_2,x_3,x_9,x_{10}$) in the first iterations while preserving accurate $Q^2$ values. For $\lambda>2$, since the additive GP is not able to capture the non-additive behavior, the performance of MaxMod decreases. Next, we show a table containing our findings.
> >
> > |$\lambda$|Sobol index $x_D$|active dimensions|knots per dimension|$Q^2$ [%]|
> > |-|-|-|-|-|
> > |0|1.7$\times$10$^{-5}$|(2,1,3)|(5,5,3)|99.8|
> > |0.5|0.02|(2,1,3,10,9)|(5,5,3,2,2)|99.2|
> > |1|0.08|(1,2,3,9,10,5)|(5,4,3,2,2)|97.6|
> > |1.5|0.15|(1,2,3,10,9)|(5,4,3,2,2)|95.5|
> > |1.7|0.18|(1,2,10,3,9,5)|(5,4,2,3,2,2)|94.7|
> >
> > We will reinforce the discussion on the non-additive nature of the underlying function of the flood application. We will also add a new appendix reporting the results of the aforementioned synthetic example.

---

> > > ### Comment · Reviewer_cQK3 · 2022-08-07
> > > **Response to authors**
> > >
> > > The response has addressed my main concerns regarding the paper. I don’t have any significant concerns remaining and am happy to raise my rating from "weak accept" to “accept”.

---

> > > > ### Author Response · Authors · 2022-08-08
> > > > **Second response to Reviewer cQK3**
> > > >
> > > > Dear Reviewer,
> > > >
> > > > Thank you for updating your rating from "weak accept" to “accept”. We appreciate it.
> > > >
> > > > The author(s)

---

### Official Review · Reviewer_RpmA · 2022-07-11

**Rating:** 6
**Confidence:** 3
**Soundness:** 3 good
**Presentation:** 3 good
**Contribution:** 3 good

**Summary:**

The authors propose an additive Gaussian process with support for monotonicity constraints and scalable to high dimensions. In particular, a sequential dimension reduction algorithm is presented which can identify the active input dimensions. The author also provided illustrative results on real-life data.



**Questions:**

1. Are the kernel parameters inferred jointly with other parameters in Eq. (11)?
2. What are the desiderata for the basis functions Eq. (4)?


**Strengths And Weaknesses:**

[Strengths]
1. The paper is well-written. The proposed GP framework is an interesting generalization of additive GPs.
2. The technical derivation looks solid, and the kernel of the proposed GP has a simple analytical form.
3. The authors also validated the method on a Vienne river flood data where the proposed MaxMod algorithm correctly activated the relevant input variables.

[Weaknesses]
1. It's not clear how the kernel parameters are estimated. In Eq. (11), the objective is given by a constrained quadratic optimization. However, this objective could be nonconvex in terms of the kernel parameters. Are there efficient solvers?
2. The proposed GP framework seems to rely on particular basis functions (Eq. (4)) which have domain [0, 1]. Does this domain impose limitations on the applicability of the GP framework? What are the desiderata for these basis functions?

---

> ### Author Response · Authors · 2022-07-29
> **Response to reviewer RpmA**
>
> We are grateful to the reviewer for the careful reading of the paper, and for the overall positive evaluation. Below are our responses to the reviewer's comments and questions.
>
> ---
>
> **On the estimation of the kernel parameters:**
> The kernel parameters are estimated via standard maximum likelihood once the GP framework is established. This is an intermediate step before solving the optimization problem in Equation (11). By fixing the kernel parameters, then (11) is convex and can be solved via quadratic programming. In particular, we use the function "solve.QP" from the R package "quadprog" (reference [21] in the paper).
>
> Although we briefly detailed in the 2D illustration that the kernel parameters are estimated via maximum likelihood, we will add a more general sentence to avoid ambiguity. We will also clarify that (11) is convex, once the kernel parameters are fixed and estimated.
>
> **On the definition of the hat basis functions:**
> The compact input domain $[0, 1]$ is considered to simplify theoretical statements and numerical implementations. In practice, one-dimensional bounded domains can be transformed into the domain $[0, 1]$ before applying the GP framework. Therefore, we argue that considering the domain $[0,1]$, for each of the input variables, does not represent a limitation on the applicability of the methodology.
>
> The hat basis functions are chosen to ensure the constraints everywhere by imposing them only at the knots (see equivalence in (6)). This is a crucial property in applications where responses satisfy physical constraints such as monotonicity or convexity. To the best of our knowledge, the aforementioned property is not entirely fulfilled by many other Bayesian or frequentist approaches from the state-of-the-art, or by considering other types of basis functions in our framework (e.g. Gaussian functions).

---

### Official Review · Reviewer_CWsg · 2022-07-11

**Rating:** 4
**Confidence:** 5
**Soundness:** 4 excellent
**Presentation:** 3 good
**Contribution:** 2 fair

**Summary:**

This work develops an algorithm for GP inference with constraints through a novel analytical update rule for an additive variant of GP inference. An active set technique for which model points to retain is developed based on Euclidean subspace projections. Numerical results illuminate the merits of the proposed approach.



**Questions:**

See response to previous field.

**Limitations:**

See response to previous field.

**Strengths And Weaknesses:**

Strengths:

The algorithm appears novel to my knowledge, and addresses a fundamental and open problem in additive reparameterizations of Bayesian linear regression with Gaussian processes.

The 2d example, as well as the extended numerical evaluations, provide substantive evidence that the proposed technique works well In practice for imposing constraints in this setting.


Weaknesses:

The authors should contrast their approach to imposing constraints through Bayesian regression with frequentist analogues. In particular, through a well-known link to kernel ridge regression, one can impose convex or linear constraints, which are very much related to the setting considered here. See, for instance:

A. Koppel, K. Zhang, H. Zhu, and T. M. Basar. ”Projected Stochastic Primal-Dual Method
for Constrained Online Learning with Kernels” in IEEE Trans. Signal Processing, May. 2019.

Marteau-Ferey, U., Bach, F., & Rudi, A. (2020). Non-parametric models for non-negative functions. Advances in neural information processing systems, 33, 12816-12826.

And follow-on works.

The authors also consider a way to deal with complexity bottleneck of GP posterior inference through a sequential dimensionality reduction approach with the squared-norm. Substantial complexity reduction of computing an inference is achieved through this active set approach. What I wonder is how is this related or different from a large history on active set techniques for complexity reduction of GP inference. In particular, recent works have  established specific convergence guarantees as a function of model complexity. How does the proposed technique contrast with these performance certificates? Is the squared-norm approach sufficient due to the additive linear nature, but beyond this situation one should consider metrics in distribution space (such as Hellinger or Wasserstein?)

McAllester DA (1999) "Pac-bayesian model averaging." In: Proceedings of the twelfth annual conference on Computational learning theory, pp 164–170

Burt D, Rasmussen CE, Van Der Wilk M (2019) "Rates of convergence for sparse variational gaussian process regression." In: International Conference on Machine Learning, pp 862–871

A. Koppel, H. Pradhan, and K. Rajawat. “Consistent Online Gaussian Process Regression
Without the Sample Complexity Bottleneck,” in Statistics and Computing, Springer, Sept.
2021

Such a contrast is not carefully addressed in this manuscript. Specifically, see Table 1 in the last of the preceding list of references.

The main theoretical results are of a nature of providing analytical expressions for the update rules. While this is credible and interesting, additional justification in the context of what would/would not guarantee constraint satisfaction is missing.

The numerical results do not really consider strong benchmarks for evaluating the tradeoffs in the constraint satisfaction/model fitness achieved by this method as compared with some alternatives. Moreover, it is unclear how does the active set approach appropriately capture the right statistical properties of the fully dense GP.



Overall opinion:

The algorithm is novel, and its update expressions combined with the active set approach to dimensionality reduction, ensures its efficiency. However, these aspects are only addressed heuristically, and no performance certificates are provided. Moreover, numerically no strong baseline is considered to determine whether the proposed technique actually achieves state of the art performance. For these reasons, the work is below the bar.

---

> ### Author Response · Authors · 2022-07-29
> **Response to reviewer CWsg  (Part 1/2)**
>
> We are grateful to the reviewer for the careful reading of the paper, for highlighting its innovative elements, for pointing out additional references to us, and for raising the important questions of numerical benchmarks and convergence guarantees. Below are our responses to the reviewer's comments and questions.
>
> ---
>
> **Link with existing frequentist works:** We are aware of the link between our work, based on Bayesian regression, and frequentist ones based on the minimization in an RKHS of a penalized least squared criterion, under inequality constraints. It was evocated in [6] (Remark 4) and studied by X. Bay, L. Grammont and H. Maatouk in:
>
> (a) Generalization of the Kimeldorf-Wahba correspondence for constrained interpolation. EJS, 10(1) 1580-1595, 2016
>
> (b) Constrained Optimal Smoothing and Bayesian Estimation, hal-03282857, 2021
>
> [Koppel et al 2019] can fit the scope of (b) when the loss function is quadratic, as the set of functions $f$ s.t. $G(f)< 0$ is convex (Eq. (2) in this reference). It provides a useful computational method, although the constraints are not satisfied everywhere in the space, a key property of our work. We will add a paragraph to advertise the link Bayesian/frequentist.
>
> [Marteau-Ferey et al 2020] does not directly address the monotonicity and convexity constraints that we consider, although it mentions convexity as future work. We will also discuss this in the paper.
>
> **Link with other works on complexity reduction:**
> [Burt et al 2019, Koppel et al 2021] address GP regression, with no constraints and with $n$ large. There, the Gaussian posterior is explicit and the computational cost for computing posterior moments or sampling from the posterior is $O(n^3)$. Hence, there are no challenges when $n$ is not large (otherwise, the observations are replaced by a smaller number of inducing points).
>
> In contrast, our constrained Gaussian posterior is not explicit. Hence, even when $n$ is not large, it is challenging to compute posterior moments or to sample from the posterior, due to the high dimensionality of the state space of MCMC procedures. Our work thus enables reducing this sampling problem dimension, by using additive functions that are parameterized by a minimal number of knots.
>
> Hence, our work and the ones mentioned above address different sources of computational complexity, and thus, arguably, their merits cannot directly be compared. We will add explanations of this difference in scope between our work and the above ones.
>
> **On the theoretical convergence guarantees:**
> [Burt et al 2019, Koppel et al 2021] provide convergence guarantees as a function of the model complexity. Currently, we do not tackle this point. As discussed above, our framework is different from theirs, so their guarantees or proof techniques cannot be applied in our setting. We will add a discussion of the existing related guarantees of the above references.
>
> Our work extends the MaxMod algorithm [18] to the additive setting, for which a convergence guarantee exists as the number of iterations increases. It needs a proof of about 15 pages. We believe that this convergence also holds for the additive case. Nevertheless, its proof appears to yield additional challenges since the additive case involves, in particular, different function spaces and multi-dimensional basis functions. Thus, this proof is currently an open question, which we consider addressing in future work.
>
> **Is the squared norm approach sufficient?**
> The main benefit of the squared norm is that it takes an explicit expression with a linear computational cost (Propositions 1 and 2). Using other metrics such as Hellinger and Wasserstein may not be as favorable computationally. In the non-additive setting [18], using the squared norm also enables to obtain the theoretical convergence of MaxMod, which we conjecture holds similarly in our setting, as discussed above. Hence, as things stand, we do not see limitations in using this norm in our setting.
>
> **On the guarantee of the constraints throughout MaxMod:**
> As previously discussed, our work verifies the constraints everywhere and not only in a finite set of points as in [8], or only approximatively as in [Koppel et al 2019]. This results from the piecewise linear approximation (see the theoretical foundation in [6]). This guarantee holds throughout the application of MaxMod since the model preserves the piecewise linear property in each step of the algorithm. This explanation was already in the paper and we will highlight it more.
>
> *(to be continued in the next comment)*

---

> > ### Author Response · Authors · 2022-07-29
> > **Response to reviewer CWsg (Part 2/2)**
> >
> > *(follow-up of Part 1)*
> >
> > **Numerical benchmark:**
> > A comparison between the constrained finite-dimensional GP and the unconstrained fully dense GP is already studied in [5]. There, the experiments show that predictions are outperformed by the former when the response satisfies constraints. We remark that in unconstrained regression as in [Burt et al 2019, Koppel et al 2021], one can indeed compare the full GP with its approximation, while in constrained regression, the full GP cannot be implemented (even when $n$ is small) because there is an infinite number of constraints, for instance for monotonicity.
> >
> > For a fair benchmark, we found two Bayesian works accounting for monotonicity over a finite set of points [Da Veiga and Marrel 2020; Riihimäki and Vehtari 2010], however, we were not able to make numerical comparisons. We only found codes for the second approach. We could not execute their codes either in R (via RcppOctave as suggested by the authors) or in Octave. The RcppOctave package seems obsolete since it was archived in 2017. We installed the latest checked version (2015) but we got an error related to the Octave configuration in both Windows and Ubuntu. We installed the GNU Octave but we got errors related to C compilers.
> >
> > [Koppel et al 2019] provide prediction functions that are close to being monotonic (among other constraints this reference tackles) so we could in principle make a numerical comparison with this work, but we have not found any mention of a publically available code in this reference.
> >
> > We will contact the authors of the three above references with the aim of accessing an implementation of one of the methods, in order to make a numerical comparison with our work.

---

> > ### Comment · Reviewer_CWsg · 2022-08-07
> > **On the Author's Response**
> >
> > The authors have addressed many of my concerns. However, the discussion of theoretical insights is lacking. What is it specifically about the "15 pages of proofs" from the prior reference that make it difficult to carry over to the setting considered in this work? Why is it categorically different from the more classical statistical consistency analyses of van der Vaart, and the recent work on performance certificates for sparse approximations? That this is unaddressed in the manuscript, even if there is no proof, seems a big hole to me. For this reason, I am disinclined to raise my score.

---

> > > ### Author Response · Authors · 2022-08-08
> > > **Second response to reviewer CWsg**
> > >
> > > We are grateful to the reviewer for the response. We next provide replies to their two questions.
> > >
> > > - **Why is extending the proof in the prior reference [18] categorically different from the more classical statistical consistency analyses of van der Vaart, and the recent work on performance certificates for sparse approximations?**
> > >
> > >    Let us explain first why the question of extending the proof in the prior reference [18] is categorically different from classical statistical consistency analyses, for instance in the book [Asymptotic statistics] by Van der Vaart. We point out that in [18], the data set of size $n$ is fixed, and the dimension of the approximation space increases. Thus, the arguments in [18] are deterministic, since the data set is fixed and there is no source of randomness. In contrast, the analysis in [Asymptotic statistics] is stochastic. We also remark that in [Asymptotic statistics], no chapters address monotonicity constraints (the only constraint addressed in the book is a unimodality constraint in density estimation). Finally, in the book [Asymptotic statistics], the number of observations $n$ goes to infinity, while in the case of [18] the data set (thus its size $n$) is fixed. Hence, the notion of consistency in [Asymptotic statistics] is different from the notion of convergence in [18].
> > >
> > >    Our response is the same as above for recent works on performance certificates for sparse approximations. For instance, in the convergence result given in Corollary 6.1 of the book [Statistics for High-Dimensional Data, Bühlmann and van de Geer, 2011], the analysis is stochastic, and consistency is defined in the setting where both the data size $n$ and dimension $p$ go to infinity.
> > >
> > > -  **What is it specifically about the "15 pages of proofs" from the prior reference [18] that make it difficult to carry over to the setting considered in this work?**
> > >
> > >    As we have written in our previous response, we think that the convergence proof of [18] can be extended to our setting, but it is difficult to anticipate the time and space needed to write a such proof, even in the favorable case where no unexpected obstacles arise. Let us now present the specific technical challenges.
> > >
> > >    In [18], the challenging point is to prove the convergence of a sequence $\hat{Y}_m$, $m \in \mathbb{N}$, of finite-dimensional constrained GP mode functions from $F \subset [0,1]^d$ to $\mathbb{R}$. Here $F = F_1 \times \dots \times F_d$ where $F_i \subset [0,1]$ is the closure of the set of one-dimensional knots for variable $i$. In the convergence, the data set of size $n$ is fixed and the number of basis functions $m$ goes to infinity. The basis functions are from $F$ to $\mathbb{R}$ and their $\mathrm{span}$ should be dense in the set of continuous functions from $F$ to $\mathbb{R}$.
> > >
> > >    In our additive setting, if we were to extend the convergence proof of [18], the fact that our mode function is additive does not imply that we can address the case of each dimension separately. Indeed, there is only one single quadratic optimization problem for all the $d$ unidimensional functions of the additive mode (Equation 11, where the optimization problem is not separable, because of the posterior cross-covariances between the additive unidimensional GPs, Line 113). Hence, we anticipate that if we have $m_1,\ldots,m_d$ knots for the $d$ dimensions, we would need to consider $m_1 + \dots + m_d$ basis functions from $[0,1]^d \to \mathbb{R}$, with for instance the function number $1$ using only the first variable and the function number $m_1 + 1$ using only the second variable. Then, we would argue that the $\mathrm{span}$ of these functions is dense in the set of additive functions from $F$ to $\mathbb{R}$. Here, again, $F = F_1 \times \dots \times F_d$ where $F_i \subset [0,1]$ is the closure of the set of one-dimensional knots for variable $i$.
> > >
> > >    Hence, the main difference with [18] is that we consider additive functions instead of general ones. This implies that we would need to see if all the arguments of [18] can be extended to this additive case. This includes, among others, defining the RKHS spaces for the limit additive function to the sequence of additive modes, defining an additive multilinear extension, defining the constraint and interpolation spaces for functions defined only on $F$ rather than on $[0,1]^d$, proving that the consequent extensions of $(H1,F)$ and $(H2,F)$ on page SM16 in the online supplement to [18] hold, and finally extending Lemma SM4.3 in the same supplement.
> > >
> > >    Note finally that [18] extends the proof of Theorem 3 in [6], which addresses the case of non-additive mode functions from $[0,1]^d$ to $\mathbb{R}$. We would thus also need to check that this proof can be extended to the case of additive functions from $[0,1]^d$ to $\mathbb{R}$.
> > >
> > > We will add a discussion to the manuscript on the prospect of obtaining a convergence proof, and the corresponding specific difficulties.

---

### Author Response · Authors · 2022-08-01
**Global comments (to all reviewers)**

We would like to thank all three reviewers again for their constructive feedback and the time spent reading the paper. In order to ease the reading of our individual responses, please find a short summary of them in this comment.

There were various requests for clarifications and discussions in specific locations in the paper, to which we tried to answer as precisely as possible. In addition, we noted, in particular, three comments on **(a)** our positioning compared to the recent literature, **(b)** theoretical convergence guarantees, and **(c)** potential additional numerical experiments.

**(a)** In the response to Reviewer CWsg, we have highlighted that the key benefit of our work compared to existing references is that we are guaranteed to satisfy the constraints (for instance monotonicity) exactly and everywhere in the space. Several of the related references that are discussed in the response do not achieve this for monotonicity.

**(b)** In the response to Reviewer CWsg, we have pointed out that several existing convergence guarantees actually apply to unconstrained GP regression, which presents fundamental differences to constrained GP regression as we study here (the posterior goes from explicit to non-explicit). Hence, these guarantees and their proofs are not applicable to our setting. A prospect of this paper is to prove the convergence of the suggested additive MaxMod algorithm, but this is expected to be a challenging extension of an existing proof in the non-additive setting (of about 15 pages).

**(c)** In the response to Reviewer cQK3, we have added a new numerical experiment to explore the robustness to non-additivity, which will be added in the paper. In the response to Reviewer CWsg, we discuss the main challenge to obtaining numerical benchmarks with existing methods for regression with monotonicity constraints: there do not seem to be publically available codes. We will send personal requests for code, in the aim of obtaining numerical benchmarks. Nevertheless, independently of these benchmarks, we know that our method cannot be improved on the criterion of satisfying the inequality constraints, as it does so exactly and everywhere on the space by construction.

In the individual responses, we have listed the modifications to the paper that we are happy to commit to do. They seem very feasible to implement to us, given the additional content page for the camera-ready version, the possibility to add appendix content, and the time left for the camera-ready version.

---

### Meta-Review · Area_Chair_65Ev · 2022-08-23

**Recommendation:** Accept
**Confidence:** Less certain

**Metareview:**

This paper deals with the problem of regression with an additive Gaussian process prior and a linear inequality constraint. A finite-dimensional approximation is proposed to the Gaussian process in terms of a linear combination of triangular basis functions with Gaussian weights. The weights are then estimated by solving a quadratic program or approximately sampled to handle the inequality constraints. Additionally, the authors consider the problem of variable selection and propose a forward selection method based on the difference between the posterior mode with and without the inclusion of a particular variable. The reviews were mixed, but are leaning towards acceptance.

**Award:**

No

---

### Decision · Program_Chairs · 2022-09-14

Accept